



# Long-term column-averaged greenhouse gas observations using a COCCON spectrometer at the high surface albedo site Gobabeb, Namibia

Matthias M. Frey[1,a], Frank Hase[1], Thomas Blumenstock[1], Darko Dubravica[1], Jochen Groß[1], Frank Göttsche[1], Martin Handjaba[2], Petrus Amadhila[2], Roland Mushi[2], Isamu Morino[3], Kei Shiomi[4], Mahesh Kumar Sha[5], Martine de Mazière[5], and David F. Pollard[6]

[1]Karlsruhe Institute of Technology (KIT), Institute for Meteorology and Climate Research (IMK-ASF), Karlsruhe, Germany
[2]Gobabeb Namib Research Institute, Gobabeb, Namibia
[3]National institute for Environmental Studies (NIES), Tsukuba, Japan
[4]Japan Aerospace Exploration Agency (JAXA), Tsukuba, Japan
[5]Royal Belgian Institute for Space Aeronomy (BIRA-IASB), Brussels, Belgium
[6]National Institute of Water and Atmospheric Research (NIWA), Lauder, New Zealand
[a]now at: National institute for Environmental Studies (NIES), Tsukuba, Japan

**Correspondence:** M. Frey (frey.matthias.max@nies.go.jp)

**Abstract.** In this study we present column-averaged dry-air mole fractions of $CO_2$ ($XCO_2$), $CH_4$ ($XCH_4$) and CO (XCO) from a recently established measurement site in Gobabeb, Namibia. Gobabeb is a hyperarid desert site at the sharp transition zone between the sand desert and the gravel plains, offering unique characteristics with respect to surface albedo properties. Measurements started January 2015 and are performed utilizing a ground-based Fourier transform infrared (FTIR) EM27/SUN

spectrometer of the COllaborative Carbon Column Observing Network (COCCON). Gobabeb is the first measurement site observing $XCO_2$ and $XCH_4$ on the African mainland and improves the global coverage of ground-based remote-sensing sites. In order to achieve the high level of precision and accuracy necessary for meaningful greenhouse gas observations, we performed calibration measurements for eight days between November 2015 and March 2016 with the COCCON reference EM27/SUN spectrometer operated at the Karlsruhe Institute of Technology. We derived scaling factors for $XCO_2$, $XCH_4$

and XCO with respect to the reference instrument that are close to 1.0. We compare the results obtained in Gobabeb to measurements at Reunion Island and Lauder from the Total Carbon Column Observing Network (TCCON). We choose these TCCON sites because, while 4000 km apart, the instruments at Gobabeb and Reunion Island operate at roughly the same latitude. The Lauder station is the southernmost TCCON station and functions as a background site without a pronounced $XCO_2$ seasonal cycle. We find a good agreement for the absolute Xgas values and representative diurnal variability. Together

with the absence of long term drifts this highlights the quality of the COCCON measurements. In Southern hemispheric summer we observe lower $XCO_2$ values at Gobabeb compared to the TCCON stations, likely due to the influence of the African biosphere. We performed coincident measurements with the Greenhouse Gases Observing Satellite (GOSAT), where GOSAT observed three nearby specific observation points, over the sand desert south of the station, directly over Gobabeb and over the gravel plains to the north. GOSAT H-gain $XCO_2$ and $XCH_4$ agree with the EM27/SUN measurements within the 1

$\sigma$ uncertainty limit. The number of coincidence soundings is limited, but we confirm a bias of 1.2 - 2.6 ppm between GOSAT





M-gain and H-gain $XCO_2$ soundings depending on the target point. This is in agreement with results reported by a previous study and the GOSAT validation team. We also report a bias of 5.9 - 9.8 ppb between GOSAT M-gain and H-gain $XCH_4$ measurements which is within the range given by the GOSAT validation team. Finally we use the COCCON measurements to evaluate inversion-optimized CAMS model data. For $XCO_2$ we find high biases of $0.9 \pm 0.5$ ppm for the OCO-2 assimilated

product and $1.1 \pm 0.6$ ppm for the in situ-driven product with $R^2 > 0.9$ in both cases. These biases are comparable to reported offsets between the model and TCCON data. The OCO-2 assimilated model product is able to reproduce the drawdown of $XCO_2$ observed by the COCCON instrument beginning of 2017, opposed to the in situ-optimized product. Also for $XCH_4$ the observed biases are in line with prior model comparisons with TCCON.

## 1 Introduction

In 2019, the concentrations of the most important anthropogenic greenhouse gases (GHGs), carbon dioxide ($CO_2$) and methane ($CH_4$), have risen to unprecedented values since the beginning of high-frequency observational records (Dlugokencky et al., 2019a, b). Additionally, it was stated recently that fossil $CO_2$ emissions exceeded 10 $GtCyr^{-1}$ for the first time in history (Friedlingstein et al., 2019). Precise and accurate global observations of GHGs are therefore important for the estimation of emission strengths, flux changes (Olsen and Randerson, 2004) and model evaluation. Furthermore, these measurements can

be directly used for the verification of climate mitigation actions as demanded by international treaties, e.g. the Paris COP21 agreement (https://unfccc.int/files/essential_background/convention/application/pdf/english_paris_agreement.pdf, last access: 15 October 2020).

Satellites like the Greenhouse Gases Observing Satellite (GOSAT) (Kuze et al., 2009; Morino et al., 2011; Yoshida et al., 2013), Orbiting Carbon Observatory-2 (OCO-2) (Frankenberg et al., 2015; Crisp et al., 2017; Eldering et al., 2017), Orbiting

Carbon Observatory-3 (OCO-3) (Eldering et al., 2019), SENTINEL5-Precursor (S5P) (Veefkind et al., 2012) or Greenhouse Gases Observing Satellite-2 (GOSAT-2) (Suto et al., 2020) are well suited candidates for this task as they retrieve total column abundances of atmospheric GHGs on a global scale. However, current satellites, while offering quasi-global spatial coverage, have coarse temporal resolution. The OCO-2 repeat cycle is 16 days, the GOSAT-2 repeat cycle is 6 days. S5P offers daily global coverage of $CH_4$ and CO. However, the measurements are mostly around local noon time. Future geostationary satel-

lites will likely help to overcome this shortcoming (Moore III et al., 2018; Nivitanont et al., 2019). Due to the fact that satellites measure backscattered sunlight from the surface of the earth and its atmosphere, retrievals of GHGs are complicated and biases can easily occur which need to be recognized and - if possible - corrected. Therefore satellite measurements are commonly validated against ground-based remote-sensing instruments as these measurements are not influenced by surface albedo effects and only minimally affected by aerosols (Dils et al., 2014; Wunch et al., 2017). The Total Carbon Column Observing Net-

work (TCCON) is a ground-based network retrieving total columns of GHGs with reference precision and accuracy utilizing high-resolution solar-viewing Fourier transform infrared (FTIR) spectrometers (Wunch et al., 2011; Washenfelder et al., 2006). TCCON is the reference instrument and primary validation source for current satellites (Inoue et al., 2016; Wu et al., 2018; Borsdorff et al., 2018).





Recently, in an effort to further improve the global coverage of ground-based observations, the COllaborative Carbon Column
Observing Network (COCCON) was established (Frey et al., 2019). This network employs compact, portable FTIR spectrom-
eters. The spectrometers used have been developed by KIT in cooperation with Bruker (Gisi et al., 2012; Hase et al., 2016) and
are commercially available since 2014 (type designation EM27/SUN spectrometer). While lately a COCCON spectrometer
was used in combination with two TCCON instruments to validate OCO-2 (Jacobs et al., 2020), to study boreal forests (Tu
et al., 2020) and recently Velazco et al. (2019) performed a campaign to validate GOSAT in central Australia, until now the
major activity of the emerging network was to create the capability of permanent COCCON measurements at remote sites as
a supplement of the existing TCCON stations by developing the procedures for ensuring proper calibration and by providing
the required evidence of the long-term stability of the EM27/SUN spectrometer (Frey et al., 2015, 2019; Sha et al., 2020).
Tasks which can be accomplished by performing differential measurements using several spectrometers which can be cali-
brated side-by-side in the framework of campaigns are easier to achieve. Many successful campaigns for quantifying GHG
emission strengths from regions of interest, as cities, coal mines, large dairy farms, etc., by arranging several spectrometers
have been performed successfully using EM27/SUN spectrometers in the recent past (Hase et al., 2015; Vogel et al., 2019;
Makarova et al., 2020; Viatte et al., 2017; Kille et al., 2019; Butz et al., 2017; Luther et al., 2019). In this work we introduce a
COCCON station in Gobabeb, Namibia, where measurements are being conducted since January 2015. The remainder of this
paper is structured as follows. In section 2 we describe the measurement site, used instrumentation and data analysis, focusing
on the COCCON EM27/SUN spectrometer. In section 3 we present the measurement results obtained over the last four years.
In section 4 a comparison with respect to TCCON stations at Reunion Island and Lauder is conducted to illustrate the feasibil-
ity of our results. Additionally, the COCCON instrument is used to validate specific target mode observations from GOSAT,
confirming a previously reported bias between GOSAT M-gain and H-gain soundings for $XCO_2$ (Velazco et al., 2019), and
for the first time also reporting a bias in $XCH_4$ for the different gain settings. We also compare our measurements to CAMS
inversion-optimized model data. In section 5 the results are discussed and an outlook for further studies is given.

## 2 Gobabeb site description, instrumentation and data analysis

### 2.1 Gobabeb site description

In 2015, we installed an EM27/SUN spectrometer of the COCCON network at the Gobabeb Namib Research Institute in
Namibia (23.561°S, 15.042°E, 410 m a.s.l.), see inset of Fig. 1. Gobabeb is located at the center of the hyperarid Namib desert.
Moreover, Gobabeb is positioned next to the Kuiseb river, which marks the sharp transition zone between the gravel plains
to the north and the sand desert to the south of the station, see Fig. 1. Gobabeb is situated 60 km east of the Atlantic ocean
and the site is approximately 80 km southeast of the closest town, Walvis Bay, with a population of about 50000. The site
is uninfluenced by nearby local emission sources of GHGs. Southwesterly winds prevail during austral summer, whereas in
winter easterly winds are predominant. The maximum temperature in summer can exceed 40 °C. Gobabeb is a high albedo
station, together with the changing terrain this results in unique site characteristics desirable especially for satellite validation
studies.



## 2.2 Description and history of the COCCON spectrometer operated at Gobabeb

The EM27/SUN spectrometer as used by COCCON has been described in great detail in the works of Gisi et al. (2012), Frey et al. (2015) and Hase et al. (2016). As a concise summary, the EM27/SUN is a solar-viewing Fourier transform infrared (FTIR)
spectrometer measuring in the near infrared spectral range (5500 - 11000 $cm^{-1}$) with a spectral resolution of 0.5 $cm^{-1}$. One measurement takes 58 s and consists of 10 individual double-sided scans. This allows the retrieval of total column abundances, $VC_{gas}$, of the target gases $O_2$, $CO_2$, $CH_4$ and $H_2O$. In 2018 the spectrometer used in this study was upgraded in Karlsruhe and a second, extended room temperature (RT) InGaAs detector (4000 - 5500 $cm^{-1}$) was added, allowing the detection of CO. During this service at KIT, the gold coating of the tracker mirrors was found to be degraded and therefore was removed
(the mirror substrate is aluminium, so the operation was continued with aluminium mirrors since then). Finally, the mechanical parts of the solar tracker attached to the spectrometer was serviced, as the very fine wind-blown dust particles tend to enter the motor stages during longer operation in the desert.

The retrieved total column abundances of the trace gases are converted into column-averaged dry air mole fractions (DMFs), where the DMF of a gas is denoted $X_{gas} = \frac{VC_{gas}}{VC_{O_2}} \times 0.2095$. Here, both the column amounts of the target gas and $O_2$ are
derived from the same spectroscopic observation reducing several potential error sources (Wunch et al., 2010). Furthermore, the dependence on the ground pressure is reduced improving comparability between different sites. A sensitive measure of the stability of a spectrometer is the column averaged amount of dry air ($X_{air}$) because for $X_{air}$ there is no compensation of possible instrumental problems, in contrast to $X_{gas}$, where errors can partially cancel out. $X_{air}$ compares the measured oxygen column ($VC_{O_2}$) with surface pressure measurements ($P_S$):

$$X_{air} = \frac{g}{P_S} \cdot \left( \frac{VC_{O_2} \cdot \overline{\mu}}{0.2095} + VC_{H_2O} \cdot \mu_{H_2O} \right) \tag{1}$$

Here $\overline{\mu}$ and $\mu_{H_2O}$ denote the molecular masses of dry air and water vapour, respectively, $g$ is the column averaged gravitational acceleration and $VC_{H_2O}$ is the total column of water vapour. The correction with $VC_{H_2O}$ is necessary as the surface pressure instruments measure the pressure of the total air column, including water vapour. Sudden changes in $X_{air}$ indicate instrumental problems, e.g. errors with the surface pressure, pointing errors, timing errors or changes in the optical alignment
of the instrument.

Frey et al. (2019) present a comprehensive characterization for EM27/SUN spectrometers used by the COCCON network, which included the instrument serial number 51 deployed in Gobabeb. In short, the instrumental line shape (ILS) of the EM27/SUN was optimized and characterized using open-path measurements as described in Frey et al. (2015), using version 14.5 of the LINEFIT retrieval software (Hase et al., 1999). Other detrimental effects, for example double-passing or channeling,
were corrected if found. For more details see section 4.2 of Frey (2018). After this initial check in December 2014 side-by-side measurements with the reference EM27/SUN and the nearby TCCON instrument were performed on the observation platform of the Institute for Meteorology and Climate Research (IMK-ASF) at the Karlsruhe Institute of Technology (KIT), Campus North (CN) near Karlsruhe (49.100° N, 8.439° E, 133 m a.s.l.). These measurements took place from November 2015 to March 2016 and once more in 2018 and 2019 in order to trace the results to TCCON (and thereby the WMO scale). This rigorous



calibration routine is necessary in order to fulfill the high precision and accuracy requirements for GHG measurements. After
the initial alignment check, no realignment was performed during the whole observation period.

The data analysis is performed differently from Frey et al. (2019). Spectra are generated from the raw interferograms (IFGs)
using a FORTRAN 2003 preprocessing tool developed in the framework of the COCCON-PROCEEDS project and extensions
(http://www.imk-asf.kit.edu/english/COCCON.php) of the European Space Agency (ESA). The IFGs are read from the OPUS

file, the solar position is calculated, a correction for direct current (DC) fluctuations following Keppel-Aleks et al. (2007) is
performed, the IFGs are truncated to the nominal resolution of 0.5 $cm^{-1}$, a numerical apodization function is applied and a
fast Fourier transformation including a phase correction routine and resampling scheme is implemented. Several quality filters
are applied, for example requiring a minimum DC level, and restricting the tolerable DC variation in the IFG or the centerburst
location in the IFG.

For the retrieval of the EM27/SUN spectra we do not use the PROFFIT 9.6 retrieval algorithm (Schneider and Hase, 2009; Kiel
et al., 2016; Chen et al., 2016). Here we use the recently developed non-linear least-squares PROFFAST retrieval algorithm
which fits atmospheric spectra by scaling a priori trace gas profiles. PROFFAST is a new efficient line-by-line forward model
and retrieval code dedicated for COCCON data analysis. It is a source-open code accessible without restrictions and is designed
to be numerically efficient and simple to use. Evaluation of data quality achieved with a COCCON spectrometer operated in

Finland including the PROFFAST data analysis chain has been investigated in the framework of ESA's FRM4GHG project and
results are reported by Sha et al. (2019). The analysis of 4 years of Gobabeb data consisting of around 120000 spectra took
about 40 h, which is approximately 30 times faster than the previously used PROFFIT 9.6 retrieval algorithm. In order to be
consistent with TCCON, the GGG2014 generated a priori files (Wunch et al., 2015) are used as a priori profiles, for trace gases
as well as for temperature and pressure. The ground pressure was recorded using a MHB-382SD data logger with a pressure

accuracy of 3 hPa (> 1000 hPa) or 2 hPa (< 1000 hPa). We use the spectroscopic line lists and retrieval windows as described
in Frey et al. (2019). The resulting $XCO_2$ and $XCH_4$ products are bias-corrected with respect to TCCON based on long-term
comparisons between COCCON data products analysed with PROFFAST and official TCCON data products from Karlsruhe
(2014 - ongoing) and Sodankyla (2017 - 2019). In the future it is planned to incorporate comparisons from additional stations
to improve the basis of the bias-correction. For $X_{air}$ a scaling factor of 0.9737 is derived from the long-term observations

performed in Karlsruhe and Sodankylä centering the $X_{air}$ data around 1.

## 2.3 TCCON Reunion Island and Lauder

Measurement procedures and data analysis at both sites follow TCCON protocol (Wunch et al., 2011) using the GGG2014
software package (Wunch et al., 2015). As required by TCCON, the instrumentation consists of a high-resolution FTIR spec-
trometer, model BRUKER IFS 125HR, which offers a maximum spectral resolution of 0.0035 $cm^{-1}$. The instrument is housed

inside a temperature-controlled building. The TCCON station in Reunion Island, France (20.901°S, 55.485°E, 87 m a.s.l.) is
located on the university campus of the Université de La Réunion in St. Denis, approximately 2000 km east of the African main-
land. The data are available via De Mazière et al. (2017). The TCCON station at Lauder, New Zealand (45.038°S, 169.684°E,
370 m a.s.l.) is situated in a sparsely populated environment on the South Island of New Zealand (Pollard et al., 2017). The





data are available via Sherlock et al. (2014); Pollard et al. (2019). In October 2018 a new TCCON instrument was installed at

Lauder. For this study we combine the data sets of both spectrometers and for the overlap period (October 2018) we use the

data from the old TCCON instrument.

### 2.4   GOSAT specific target observations

A detailed description of the GOSAT instrumental features and data analysis is given in Kuze et al. (2009) and Yoshida et al.

(2013). GOSAT detects shortwave-infrared radiation in three narrow bands (0.76, 1.6 and 2.0 $\mu$m) with a resolution of 0.2

cm$^{-1}$. Additionally it is equipped with a sensor measuring in the thermal infrared range. The TANSO-FTS footprint diameter

is about 10.5 km at sea level. The nominal single-scan acquisition time is 4 s. For this study the GOSAT FTS Short Wave

InfraRed (SWIR) Level 2 data version V02.81 from NIES is used. The satellite is flying at an altitude of 666 km with a

repeat cycle of 3 days. Starting May 2016, GOSAT performed specific target mode observations over Gobabeb by performing

observations at three distinct points, see Fig. 1. Directly at the Gobabeb COCCON site, approximately 10 km north east over

the gravel plains and around 10 km south west over the sand desert. These points were chosen because of their different surface

reflectance in order to study the sensitivity of the GOSAT retrieval with respect to the surface albedo. The satellite performed

measurements with different gain settings, M-gain and H-gain. M-gain soundings are generally performed over surfaces that

are bright in the near infrared. For M-gain observations other validation sites with ground-based FTIR measurements are sparse

(Yoshida et al., 2013; Velazco et al., 2019).

### 170   2.5   CAMS global CO$_2$ and CH$_4$ atmospheric inversion products

The CAMS model has been described previously in great detail, e.g. (Agustí-Panareda et al., 2014; Massart et al., 2016; Inness

et al., 2019). Here we utilize the CAMS global inversion-optimized column-averaged dry air mole fractions for CO$_2$ and CH$_4$.

For CO$_2$, we use an inversion product FT19r1 (Chevallier, 2020a) assimilating OCO-2 satellite observations (O'Dell et al.,

2018; Kiel et al., 2019) as well as an in situ driven inversion product v18r3 (Chevallier, 2019). More details can be found in

Chevallier et al. (2019); Chevallier (2020b). For CH$_4$, an inversion product v18r1s assimilating a combination of surface and

GOSAT satellite observations (Detmers and Hasekamp, 2016) as well as one product v18r1s using only surface observations

are analyzed (Segers and Houweling, 2020a). A description of the inversion procedure together with comparisons against

independent observational data sets is given in Segers and Houweling (2020b).

## 3   Measurement results

### 180   3.1   Side-by-side measurements at Karlsruhe

ILS measurements were carried out seven times since December 2014. The modulation efficiency (ME) at maximum optical

path difference (MOPD) ranges between 0.979 and 0.986 with a mean value of 0.983 and a standard deviation of 0.002. The

mean phase error is 0.0019 $\pm$ 0.0003. No drift is apparent and the ILS is stable. The spread in the ME is in good agreement





with the error budget of 0.003 given in Frey et al. (2019). This high instrumental stability is remarkable and not self-evident.

Between the measurements the EM27/SUN was shipped from Karlsruhe to Gobabeb, including airlift and transport by car on bumpy gravel roads.

Between November 2015 and March 2016 side-by-side comparison measurements were conducted on eight days together with the reference EM27/SUN to derive calibration factors for the different trace gases for this spectrometer and thereby removing possible instrument-dependent biases. Some data had to be filtered out due to different reasons. Because most measurements

were performed during winter, the solar elevation was low, which sometimes led to a partially obstructed view due to railings and a metal frame on the terrace where the observations took place. In the morning the first measurements were omitted due to unusually high scatter caused by the quickly changing temperature of the not frequency-stabilized HeNe laser, as already reported by Gisi et al. (2012). In rare cases, the tracking software failed, resulting in corrupted spectra, that were also filtered out. For this analysis only observations from the two instruments performed within one minute and solar zenith angles (SZAs)

below $85°$ are taken into account, resulting in 1209 coincident measurements. The results are shown in Fig. 2. The derived instrument-specific calibration factors are $1.0002 \pm 0.0003$ for $XCO_2$, $1.0005 \pm 0.0004$ for $XCH_4$, $1.0011 \pm 0.0029$ for $XH_2O$ and $0.9995 \pm 0.0005$ $X_{air}$ between the reference instrument and the instrument deployed in Namibia. Although the scaling factors are close to nominal for all species, to avoid biases due to instrumental differences these calibration factors are taken into account in the analysis of the Namibia data set.

Additional side-by-side measurements were performed in February and March 2018 after the instrument came back from Namibia as well as between November 2018 and February 2019 after the dual channel upgrade and mirror exchange. The combined results are shown in Appendix A. A slight variation in the calibration factors is detectable, for $XCH_4$ and $X_{air}$ the change is significant at the $1 \sigma$ level. The numeric values for the scaling factors are $1.0004 \pm 0.0004$ for $XCO_2$, $0.9989 \pm 0.0004$ for $XCH_4$, $0.9988 \pm 0.0016$ for $XH_2O$ and $1.0031 \pm 0.0007$ for $X_{air}$. For the period between November 2018 and

February 2019 we also derive a calibration factor of $0.9940 \pm 0.0050$ for XCO. As the bias between the calibration factors obtained during the two side-by-side measurement periods is within 0.1 ppm for $XCO_2$, 3 ppb for $XCH_4$ and 3 ppm for $XH_2O$, for the analysis of the Namibia data we will only use the mean calibration factors derived from these observation periods.

## 3.2 Gobabeb Xgas time series

For the subsequent analysis only observations with SZAs not exceeding $80°$ are taken into account, resulting in 113049 indi-

vidual measurements on 319 days between 2015 and 2019. In Fig. 3 we present the $XCO_2$, $XCH_4$, XCO, $XH_2O$ and $X_{air}$ retrieval results from the COCCON Gobabeb observations. For better visibility, daily mean values are shown. Error bars denote the $1 \sigma$ standard deviation of the daily mean values. For $XCO_2$, the underlying trend of about 2 ppm / year can be seen. Correspondingly, a daily minimum value was observed at the beginning of the measurements on 24 January 2015 with $394.3 \pm 0.2$ ppm and the maximum daily value was observed on 15 October 2019 ($410.6 \pm 0.2$ ppm). A seasonal cycle is also detectable,

with a peak-to-peak amplitude of 5.3 ppm in 2017. Here it is calculated as the difference between the maximum monthly mean of $404.0 \pm 1.1$ ppm in September and the minimum monthly mean of $398.0 \pm 0.5$ in March. This amplitude is higher than observed in other southern hemisphere TCCON stations in Australia and New Zealand (Deutscher et al., 2014), owing to a





rather sharp drawdown of $XCO_2$ in February and March 2017. However, this is probably a real signal as the impact of the biosphere in Africa might lead to a larger seasonal cycle in Gobabeb. Also Olsen and Randerson predict a rather prominent

$XCO_2$ seasonal cycle on the order of 5 ppm in southern Africa, see Figure 5 of Olsen and Randerson (2004). For $XCH_4$, daily mean values range between $1759 \pm 1$ ppb (2 June 2015) and $1828 \pm 1$ ppb (25 June 2019). The trend is roughly 0.01 ppm / year. The $XCH_4$ seasonal cycle has lowest values in southern hemispheric summer (January 2017: $1783 \pm 5$ ppb) and highest values throughout winter and early spring (September 2017: $1808 \pm 5$ ppb) resulting in a peak-to-peak amplitude of 25 ppb. Regarding XCO, the time series is limited to 2019 due to the fact that the dual channel upgrade was only performed in 2018.

At this point, it can already be seen that this site observes highly variable amounts of carbon monoxide, ranging from very clean background conditions with daily mean XCO values as low as $49 \pm 1$ ppb (16 April 2019) to elevated results of up to $131 \pm 9$ ppb (4 September 2019). $XH_2O$ is very low during large parts of the year, as expected for a desert site. The lowest value was reached on 29 June 2015 ($357 \pm 10$ ppm). During late southern hemispheric summer and early spring, $XH_2O$ can reach up to several thousand ppm. As mentioned in section 2.2, $X_{air}$ is an important parameter to monitor the instrumental

stability. For the whole time series, daily $X_{air}$ results are stable within 1 %. No apparent drift of $X_{air}$ is visible during the four years of measurements performed at the COCCON Gobabeb station.

## 4   Gobabeb data comparisons

### 4.1   TCCON Reunion Island and Lauder

In this section we compare the results obtained in Gobabeb with results from the TCCON stations at Reunion Island and

Lauder. Although this is not a side-by-side comparison, Reunion Island as the second closest TCCON station is approximately 4000 km east of Gobabeb, this comparison will give us a measure of the feasibility of our results. The observations should be comparable qualitatively as the variation of $XCO_2$ is relatively low in the southern hemisphere compared to the northern hemisphere (Olsen and Randerson, 2004). Moreover, Gobabeb (24°S) and Reunion Island (21°S) are roughly at the same latitude. The TCCON station Ascension Island is slightly closer to Gobabeb with a distance of approximately 3600 km, but

the latitudional difference is larger. Due to the latitudional gradient in $XCH_4$, we therefore chose to compare our COCCON measurements to Reunion Island rather than Ascension Island. Lauder is the southernmost TCCON station and functions as a background site without a pronounced $XCO_2$ seasonal cycle.

Daily mean $XCO_2$, $XCH_4$, XCO and $XH_2O$ results are shown in Fig. 4 from COCCON Gobabeb (blue dots), TCCON Reunion Island (black dots) and TCCON Lauder (red dots) stations. Error bars denote the 1 $\sigma$ standard deviation of the daily

mean values. For $XCO_2$ we see a good agreement between the sites, given the fact that they are spatially far apart. The annual increase of $XCO_2$ is similar for all stations. For Reunion Island and Lauder, no pronounced seasonal cycle is visible. Most prominent difference is the sharp decrease of $XCO_2$ at Gobabeb beginning of 2017, most pronounced in March. This is not seen for the TCCON data at the two other sites. As discussed in the previous section, this is probably due to the impact of the African biosphere to the measurements in Gobabeb. To a smaller extent this difference can also be seen at the beginning

of 2018. Despite the similarities, at the beginning of 2018 and then at the end of 2019 it can also be seen that the Reunion





Island values somewhat diverge from the Gobabeb and Lauder values. $XCH_4$ at Gobabeb and Reunion sites is similar, with lower absolute values at Lauder. The annual increase as well as the seasonal variability are similar at all sites. Opposed to $XCO_2$, there is no conspicuous difference between the data sets at the beginning of 2017. For XCO, the sites do not have a long observation overlap, it seems that the variability is slightly larger in the COCCON data. Regarding $XH_2O$, the seasonality

is similar between the sites, with highest values at Reunion Island throughout the year.

In order to affirm that the drawdown of $XCO_2$ at the beginning of 2017 at the Gobabeb station is due to the influence of the African biosphere, we show 10-day backward trajectory ensemble simulations from the National Oceanic and Atmospheric Administration (NOAA) HYSPLIT model (Stein et al., 2016) for 16 February 2017, the day with the lowest $XCO_2$ values in 2017. Initial 3-hourly meteorological input data is provided by the NCEP Global Data Assimilation System (GDAS) model on

a 1 degree latitude-longitude grid. The end point of the trajectory analysis is chosen at a height of 5000 m above ground level. We choose this height because in section 4.3 a comparison between COCCON data with CAMS model data shows that the CAMS model version assimilating total column data reproduces the $XCO_2$ drawdown, in contrast to the version assimilating in situ data. Therefore we think that the drawdown is driven by low concentrations of $CO_2$ in the higher layers in the atmosphere rather than in the atmospheric boundary layer. Backward trajectories for Gobabeb are depicted in Fig. 5. All trajectories exhibit

a long dwell time over the African continent, corroborating the conjecture that the low $XCO_2$ values at Gobabeb are due to the influence of the African biosphere. In contrast, the backward trajectories for Reunion Island shown in Fig. 6 dwell almost exclusively over the ocean.

In a next step, we show correlation plots for the COCCON site with respect to the TCCON sites for $XCO_2$ and $XCH_4$ in Fig. 7 and 8. Error bars denote the 1 $\sigma$ standard deviation (STD) of the daily mean values. The colorbar denotes the measurement

date. Focussing first on the comparison between Gobabeb and Reunion Island in Fig. 7, we find an agreement within one standard deviation of the averaged daily mean values for both gases. For $XCO_2$ a scaling factor of $1.0027 \pm 0.0028$ and a correlation coefficient $R^2$ of 0.911 are derived. For $XCH_4$ the scaling factor is $1.0028 \pm 0.0045$ and $R^2$ of 0.670. Bias and STD in absolute values are given in Table 1. Despite this good agreement, especially for $XCO_2$ there is some divergence between the data before and after 2018, corresponding to larger scatter in the TCCON Reunion Island data set, as can be seen by the

larger error bars for the 2018 and 2019 data. For 2018 the reason for the increased scatter was continued mirror degradation as a result of sea salt deposition from the ocean. In Fig. 8 we see an excellent agreement between the COCCON Gobabeb and TCCON Lauder data for $XCO_2$ with a scaling factor of $0.9990 \pm 0.0027$ and a correlation coefficient $R^2$ of 0.906. The only discernible anomaly are the lower COCCON values beginning of 2017, which is also seen in the time series in Fig. 4. Otherwise, no temporal drift between the two data sets is apparent. For $XCH_4$ a scaling factor of $0.9800 \pm 0.0060$ with $R^2 =$

0.556 is found. The large bias is to be expected due to the latitudinal gradient in atmospheric methane concentrations (Saeki et al., 2013).

Next, we examine several diurnal cycles for five days between Gobabeb and Reunion Island, one day each year, where data is available for both sites. The results for $XCO_2$ and $XCH_4$ are shown in Fig. 9, COCCON measurements are shown as blue dots, TCCON measurements as black dots. In contrast to other graphs, here we show local time data, for better comparability of the

diurnal cycles. For $XCO_2$, the diurnal curvature for both COCCON and TCCON is relatively flat, however a slight parabola



shape is discernible. For southern hemispheric summer, compared to TCCON Reunion COCCON Gobabeb values are slightly lower as was already seen in the time series analysis. XCH$_4$ diurnal variations are similar for both sites, also the absolute values are in perfect agreement. A common feature for both data sets is the apparent parabola shape on most days. This is probably the result of a combination of non-perfect a priori profiles, residual airmass dependency and intraday changes of atmospheric

temperature. In the next version of the TCCON trace gas retrieval algorithm, updated a priori profiles will be used that will help to further reduce these unwanted effects. For 13 July 2015 it seems that this effect is slightly more pronounced for the COCCON instrument. For the other days this is hard to assess as the scatter of the TCCON Reunion Island data continuously increases with time due to degrading mirror quality. This finding is true for both XCO$_2$ and XCH$_4$.

## 4.2 GOSAT validation

In this section we validate specific target mode observations from the GOSAT satellite around Gobabeb at three distinct points with different surface albedo properties against COCCON Gobabeb observations. Target mode measurements started 2016 and are ongoing. The time series of the GOSAT observations is shown in Fig. 10. Measurements over the gravel plains are displayed in red, observations directly at Gobabeb in black and measurements over the sand desert are presented in gold, with 59, 78 and 85 successful observations, respectively. In general, the agreement between GOSAT observations and COCCON

measurements is reasonable, GOSAT data seem to be slightly biased high both in XCO$_2$ and XCH$_4$. An interesting anomaly is observed in the GOSAT data, there seems to be a small decrease both in XCO$_2$ and XCH$_4$ during southern hemispheric winter, which is not observed by the COCCON instrument. For a rigorous assessment the data is too sparse however. An additional difference is that the drawdown of XCO$_2$ values beginning of 2017 is more pronounced for COCCON compared to the satellite data.

For a quantitative analysis, we analyze coincident observations between GOSAT and COCCON. To make the data sets comparable, we correct for the influence of the different a priori profiles following Rodgers and Connor (2003). We adjust the GOSAT values to the ensemble profile, which we assume to be the GGG2014 generated a priori profile. In Fig. 11 we present the XCO$_2$ and XCH$_4$ COCCON and GOSAT averaging kernels for different SZAs. Although the COCCON averaging kernels are shown for SZAs in the range of 0° and 85°, for all coincident overpasses the SZA was between 10° and 50°. Due to the

similarities of the averaging kernels, we neglect the smoothing error in the following analysis.

The number of coincident measurements with COCCON observations are 13, 18 and 20 for the three specific observation points and the chosen coincidence criteria is that COCCON observations were performed within thirty minutes of the satellite overpass. Of these coincident measurements, the vast majority occurred in 2016. The correlation graphs for these three target points are presented in Fig. 12, 13 and 14. GOSAT M-gain observations are color-coded red, while H-gain observations are

shown in blue. Error bars denote the 1 $\sigma$ standard deviation of the hourly mean values for COCCON measurements and the measurement error for the GOSAT soundings.

For the GOSAT observations over the gravel plains, only GOSAT M-gain soundings were performed. The spread of the data set is relatively large, GOSAT is biased high and we derive a scaling factor with respect to the COCCON observations of 1.0062 $\pm$ 0.0026 and 1.0044 $\pm$ 0.0039 for XCO$_2$ and XCH$_4$, where the difference is statistically significant at the 1 $\sigma$ level. This





corresponds to a high bias of $2.5 \pm 1.1$ ppm for $XCO_2$ and $7.9 \pm 7.1$ ppb for $XCH_4$. In Table 2 and Table 3 the absolute values of the GOSAT - COCCON comparison are summarized. Directly over Gobabeb GOSAT M-gain as well as H-gain soundings were performed. Between COCCON and GOSAT M-gain data we derive a scaling factor of $1.0026 \pm 0.0027$ for $XCO_2$ and $1.0018 \pm 0.0033$ for $XCH_4$, corresponding to a high bias of $1.0 \pm 1.1$ ppm for $XCO_2$ and $3.1 \pm 6.0$ ppb for $XCH_4$. For H-Gain observations we derive a scaling factor of $0.9996 \pm 0.0020$ for $XCO_2$ and $0.9984 \pm 0.0016$ for $XCH_4$, corresponding

to a low bias of $0.2 \pm 0.8$ ppm for $XCO_2$ and $2.8 \pm 2.9$ ppb for $XCH_4$. The differences between GOSAT and COCCON are not statistically different at the $1\,\sigma$ level. Over the sand desert, the GOSAT M-gain data are biased high with respect to the COCCON data with a scaling factor of $1.0068 \pm 0.0026$ for $XCO_2$ and $1.0070 \pm 0.0045$ for $XCH_4$, corresponding to a high bias of $2.7 \pm 1.1$ ppm for $XCO_2$ and $12.5 \pm 8.1$ ppb for $XCH_4$. The H-gain data are in very good agreement with the COCCON observations with a scaling factor of $1.0003 \pm 0.0008$ for $XCO_2$ and $1.0015 \pm 0.0028$ for $XCH_4$, corresponding to

a high bias of $0.1 \pm 0.3$ ppm for $XCO_2$ and a high bias of $2.7 \pm 5.1$ ppb for $XCH_4$.

Although not always statistically significant at the $1\,\sigma$ level, clear differences are discernible between the different GOSAT gain settings. This is in agreement with results reported by Velazco et al. (2019) and the GOSAT validation team. For the H-gain soundings we report a good agreement with the COCCON observations within the $1\,\sigma$ level for $XCO_2$ as well as $XCH_4$ with high correlation coefficients ($R^2 > 0.9$).

## 4.3  CAMS evaluation

As was shown in section 4.1, the COCCON measurements exhibit a small but discernible parabola shape during the day. For better comparability, we therefore only compare COCCON measurements around local noon with the CAMS model data. Although using all COCCON data instead of only noon data results in only a small bias of 0.2 ppm for $XCO_2$ and 2 ppb for $XCH_4$, we feel that this is the more consistent comparison. The resulting correlation plots for $XCO_2$ and $XCH_4$ are presented

in Fig. 15 and 16. For $XCO_2$, note that the OCO-2 assimilated data is available until 2019 and the in situ assimilated data is available until 2018. The left panel of Fig. 15 shows the OCO-2 assimilated model data. We see an excellent agreement between the two data sets with a bias of $0.9 \pm 0.5$ ppm and a correlation coefficient $R^2$ of 0.983. This offset agrees well with the bias between CAMS model and TCCON data presented in Chevallier (2020a). We do not observe an increased bias at the beginning of 2017. This means that the OCO-2 assimilated model reproduces the drawdown of $XCO_2$ seen in the COCCON time series

in Fig. 3 during this time. In contrast, we see an increased bias during the beginning of 2017 in the in situ assimilated data in the right panel. Apart from this anomaly, the agreement between the two data sets is good. The CAMS model has a high bias of $1.1 \pm 0.6$ ppm and $R^2 = 0.927$. The absolute values of the CAMS model evaluation are depicted in Table 4.

For $XCH_4$, both the combined GOSAT and in situ assimilated data as well as the in situ assimilated data are available until 2018. The GOSAT and in situ assimilated CAMS data exhibit a low bias of $2.4 \pm 8.0$ ppb, $R^2 = 0.455$. From end of 2016 to

beginning of 2017 an anomaly is discernible with higher CAMS values. This is not seen in the comparison with the in situ assimilated dataset. The anomaly corresponds to a period of increased scatter in the GOSAT and in situ assimilated CAMS timeseries itself. Therefore we attribute this anomaly to the influence of the GOSAT observations. For the in situ assimilated data we find a low bias of $5.8 \pm 4.8$ ppb and $R^2 = 0.645$. This is consistent with the low bias of CAMS with respect to TCCON





measurements in the latitude band between 20°S and 30°S of around 10 ppb, as shown in figure 17 of Segers and Houweling

(2020a).

## 5 Conclusions and Outlook

We present measurements from a new ground-based remote-sensing COCCON station in Namibia, the first FTIR site measuring GHGs on the African continent. We performed a thorough calibration scheme carried out in Karlsruhe in order to make results traceable to TCCON (and thereby the WMO scale), including ILS measurements and side-by-side comparisons with

a reference COCCON spectrometer. The results from Namibia show typical global annual increase rates for both $XCO_2$ as well as $XCH_4$. In contrast to comparable FTIR measurements in the southern hemisphere, we observe a pronounced seasonal variability for $XCO_2$ with a peak-to-peak amplitude of 5.3 ppm in 2017, in agreement with OCO-2 assimilated CAMS model data and global transport model predictions (Olsen and Randerson, 2004). As expected for a desert site, we observe very low values of $XH_2O$, with a minimum value of 357 ppm. For the whole time series, daily $X_{air}$ results are stable within 1 %. No

apparent drift of $X_{air}$ is visible during the four years of measurements performed at the COCCON Gobabeb station.

To put our results in the broader geophysical context, we compare the COCCON Namibia results to measurements from the TCCON stations Reunion Island and Lauder. Given the fact that the stations are spatially far apart, the results are in good agreement. For $XCO_2$ both TCCON Lauder (-0.4 ± 1.1 ppm) and Reunion Island (1.1 ± 1.1 ppm) show biases compared to COCCON Gobabeb within the 1 $\sigma$ uncertainty range and correlation coefficients $R^2$ > 0.9. For $XCH_4$ TCCON Reunion Island

and COCCON Gobabeb data agree within the 1 $\sigma$ uncertainty range (5.1 ± 8.1 ppb) while a large bias (-35.9 ± 10.6 ppb) is observed with respect to the Lauder data. This is a direct result of the strong latitudinal gradient in total column averaged methane concentrations. We further investigate the diurnal variations from TCCON Reunion Island and COCCON Gobabeb for $XCO_2$ and $XCH_4$. Both share a small but systematic downward parabola shape, probably the result of a combination of non-perfect a priori profiles, residual airmass dependency and intraday changes of atmospheric temperature. From a comparison of the two

data sets we also deduce that the Reunion Island data set shows increased scatter during some time periods due to the degrading mirror quality as a result of sea salt deposition from the ocean. Compared to the TCCON results, the COCCON observations are of comparable quality.

We show the usefulness of our station for satellite validation by comparing the COCCON results to GOSAT specific target mode observations at three points close to or directly at the site with different surface albedos. The satellite performed measurements

with different gain settings. Ground-based validation of the different gain settings is difficult as very few sites worldwide have the necessary surface characteristics, further supporting the importance of this new station. We find a good agreement between GOSAT H-gain and COCCON observations within the 1 $\sigma$ uncertainty range with low biases of -0.2 ± 0.8 ppm for $XCO_2$ and -2.8 ± 2.9 ppb for $XCH_4$ at Gobabeb and high biases of 0.1 ± 0.3 ppm for $XCO_2$ and 2.7 ± 5.1 ppb for $XCH_4$ over the sand desert approximately 15 kilometers south-east of the station. For M-gain soundings, GOSAT measurements are always biased

high with respect to the COCCON measurements, the differences over the gravel plains and the sand desert are statistically significant at the 1 $\sigma$ level. Thereby we show the capability of this site to validate satellite measurements for different high

albedo surfaces.

Then we evaluate the performance of the inversion-optimized CAMS model data against our ground-based COCCON data. For $XCO_2$ we find high biases of $0.9 \pm 0.5$ ppm for the OCO-2 assimilated product and $1.1 \pm 0.6$ ppm for the in situ-driven

product with $R^2 > 0.9$ in both cases. These biases are comparable to offsets between the model and TCCON data. The OCO-2 assimilated model product is able to reproduce the drawdown of $XCO_2$ beginning of 2017, as opposed to the in situ-optimized product. Also for $XCH_4$ the biases found are in line with prior model comparisons with TCCON.

With this work we show the potential of the COCCON network for satellite validation and atmospheric transport model validation. We expect that the availability of additional COCCON sites in the near future will be a great asset for future satellite and

model studies as they are easy to deploy. In the course of the ESA funded COCCON PROCEEDS project COCCON data from several sites will be made available via a web portal. We conclude that instruments from the COCCON network offer stable long-term records of GHGs in remote environments and can be used to close gaps in the global distribution of ground-based remote-sensing sites.

*Data availability.*   COCCON data will be made available in the near future through a web portal hosted at the Karlsruhe Institute of Tech-

nology. TCCON Reunion Island and Lauder data can be obtained via: https://tccondata.org, last access: 20 October 2020. The GOSAT TANSO-FTS SWIR L2 data are available from the GOSAT Data Archive Service (GDAS) at https://data2.gosat.nies.go.jp/ (GDAS, last access: 20 October 2020).

## Appendix A:  Calibration measurements Karlsruhe 2018 and 2019

In Fig. A1 we present the results from the calibration measurements performed between February 2018 and 2019.

*Author contributions.*   MF, TB and FH planned the study. MF and TB installed the COCCON station in Gobabeb with help from FH, FG and JG. RM, MH and PA performed the day to day COCCON measurements. MKS and MDM provided the TCCON Reunion Island data. DFP provided the TCCON Lauder data. DD provided the COCCON Gobabeb data. IM provided the GOSAT data. KS organized the GOSAT specific target mode observations over Gobabeb. MF performed the data analysis and wrote the paper. All authors reviewed, edited and approved the paper.

*Competing interests.*   The authors declare that they have no conflict of interest.

*Acknowledgements.*   CAMS data were generated using Copernicus Atmosphere Monitoring Service Information 2020.
The authors acknowledge support by the ACROSS research infrastructure of the Helmholtz Association of German Research Centres (HGF).



The authors acknowledge support by the MOSES research infrastructure of the HGF.

The authors acknowledge support by ESA in the framework of FRM4GHG, COCCON-PROCEEDS, COCCON-PROCEEDSII and QA4EO
projects.

The Réunion Island station is operated by the Royal Belgian Institute for Space Aeronomy with financial support since 2014 by the EU
project ICOS-Inwire and the ministerial decree for ICOS (FR/35/IC1 to FR/35/IC5) and local activities supported by LACy/UMR8105 –
Université de La Réunion.

The authors gratefully acknowledge the NOAA Air Resources Laboratory (ARL) for the provision of the HYSPLIT transport and dispersion
model used in this publication.



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





**Table 1.** This table presents the results of the comparison between the COCCON station in Gobabeb and the TCCON stations in Lauder and Reunion Island. Bias and STD are given as the mean difference and one standard deviation between the coincident daily TCCON and COCCON $XCO_2$ and $XCH_4$ values.

| Station | $XCO_2$ Bias $\pm$ STD [ppm] | $XCH_4$ Bias $\pm$ STD [ppb] | Number of coincidences |
|---|---|---|---|
| Reunion Island | $1.1 \pm 1.1$ | $5.1 \pm 8.1$ | 155 |
| Lauder | $-0.4 \pm 1.1$ | $-35.9 \pm 10.6$ | 241 |





**Table 2.** This table presents the results of the comparison between the COCCON station in Namibia and the GOSAT M-gain specific target observations. Bias and STD are given as the mean difference and one standard deviation between the coincident GOSAT and COCCON observations.

| GOSAT target point | M-gain $XCO_2$ Bias $\pm$ STD [ppm] | M-gain $XCH_4$ Bias $\pm$ STD [ppb] | Number of coincidences |
|:---:|:---:|:---:|:---:|
| Gravel plains | $2.5 \pm 1.1$ | $7.9 \pm 7.1$ | 13 |
| Gobabeb | $1.0 \pm 1.1$ | $3.1 \pm 6.0$ | 13 |
| Sand desert | $2.7 \pm 1.1$ | $12.5 \pm 8.1$ | 12 |



**Table 3.** This table presents the results of the comparison between the COCCON station in Namibia and the GOSAT H-gain specific target observations. Bias and STD are given as the mean difference and one standard deviation between the coincident GOSAT and COCCON observations.

| GOSAT target point | H-gain $XCO_2$ Bias $\pm$ STD [ppm] | H-gain $XCH_4$ Bias $\pm$ STD [ppb] | Number of coincidences |
|:---:|:---:|:---:|:---:|
| Gravel plains | - | - | 0 |
| Gobabeb | -0.2 $\pm$ 0.8 | -2.8 $\pm$ 2.9 | 5 |
| Sand desert | 0.1 $\pm$ 0.3 | 2.7 $\pm$ 5.1 | 8 |





**Table 4.** This table presents the results of the comparison between the COCCON station in Namibia and the assimilated CAMS model data. Bias and STD are given as the mean difference and one standard deviation between the coincident hourly-pooled local noon COCCON and CAMS $XCO_2$ and $XCH_4$ values.

| Assimilation data | $XCO_2$ Bias $\pm$ STD [ppm] | $XCH_4$ Bias $\pm$ STD [ppb] | Number of coincidences |
|---|---|---|---|
| OCO-2 data | $0.9 \pm 0.5$ | - | 263 |
| In situ data | $1.1 \pm 0.6$ | - | 187 |
| In situ and GOSAT data | - | $-2.4 \pm 8.0$ | 187 |
| In situ data | - | $-5.8 \pm 4.8$ | 187 |



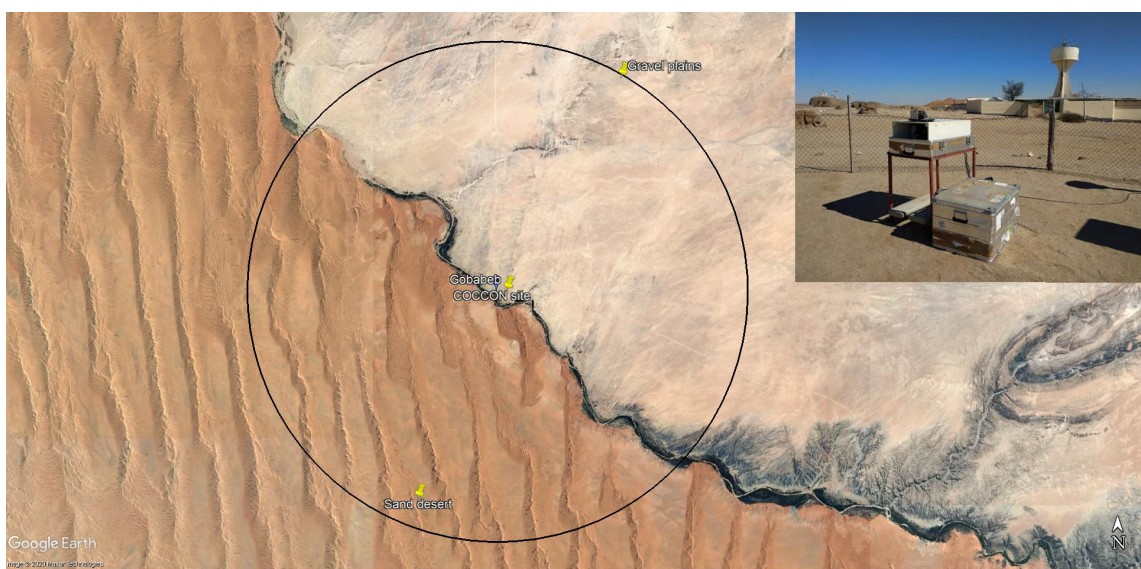

**Figure 1.** © Google earth image (Map data: © Google, Maxar technologies) of the measurement site at Gobabeb, Namibia. The blue pin denotes the position of the COCCON instrument. The yellow points show the positions of the GOSAT target observation points. A black circle with a radius of 10 km has been drawn around the COCCON site for visual reference. The inset in the upper right corner shows the EM27/SUN spectrometer at Gobabeb.

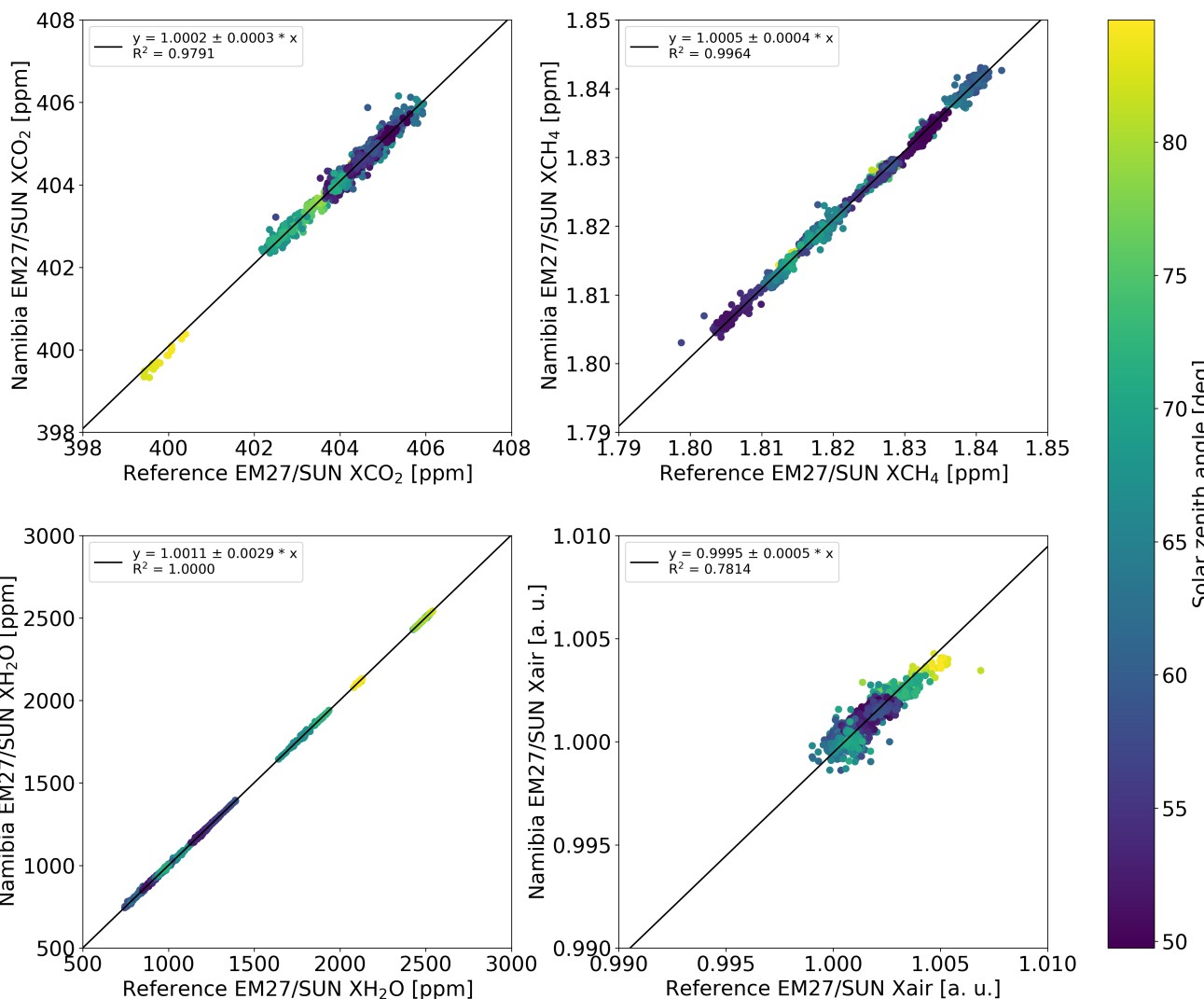

**Figure 2.** Side-by-side measurements between the reference EM27/SUN and the instrument deployed in Namibia performed between November 2015 and March 2016 in Karlsruhe. From left to right, the panels show correlation plots for $XCO_2$, $XCH_4$, $XH_2O$ and $X_{air}$. The coincident criteria is that measurements for both instruments occured within one minute. The colorbar denotes the solar zenith angle. For the analysis, only measurements with zenith angles below $85°$ are considered.

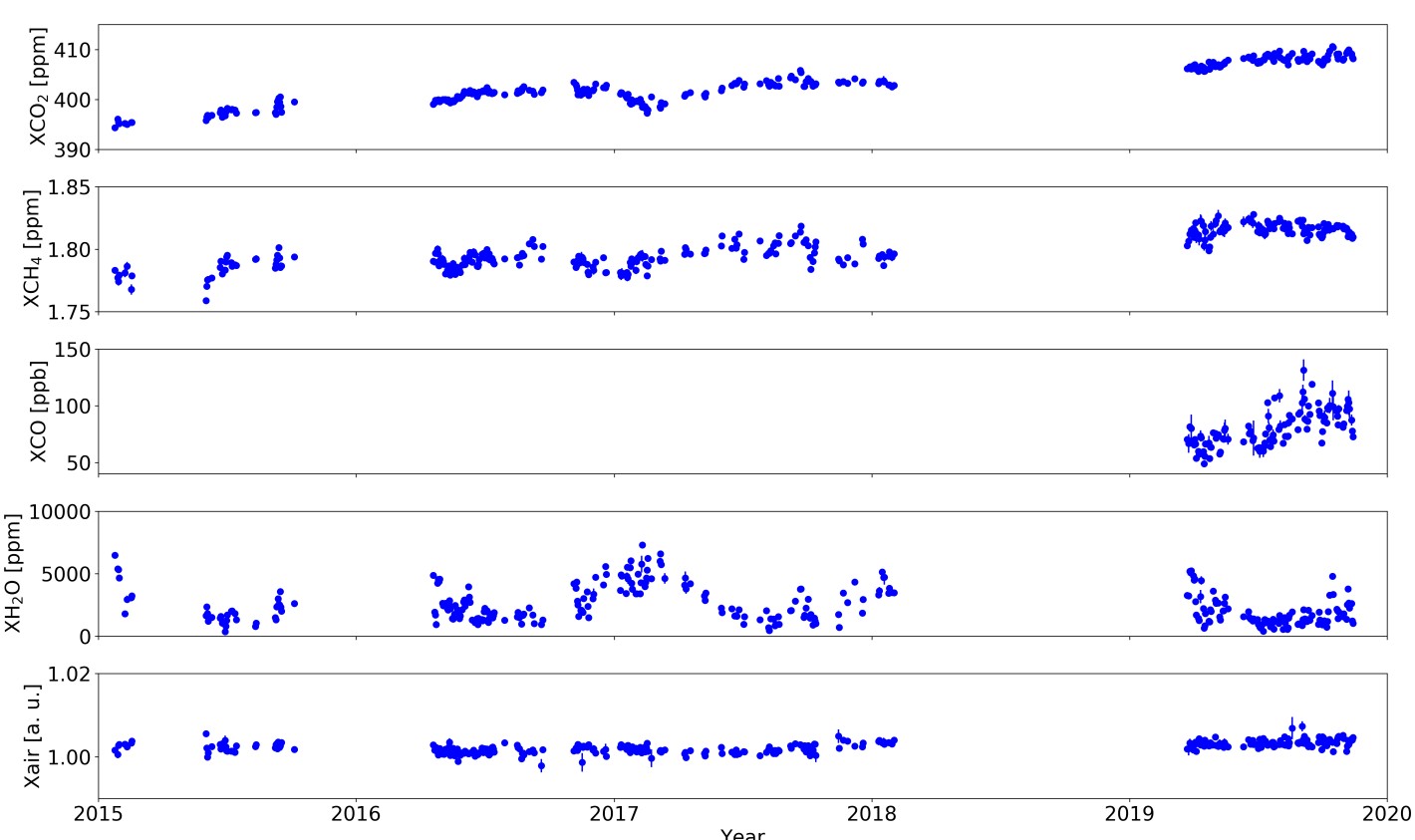

**Figure 3.** Column-averaged dry air mole fraction time series for $XCO_2$, $XCH_4$, $XCO$, $XH_2O$ and $X_{air}$ measured at the COCCON site in Gobabeb, Namibia from January 2015 until November 2019. Daily mean values are shown for better visibility. Error bars denote the $1\ \sigma$ standard deviation of the daily mean values. In 2018 the instrument was upgraded with a second channel. Therefore XCO observations only started in 2019.

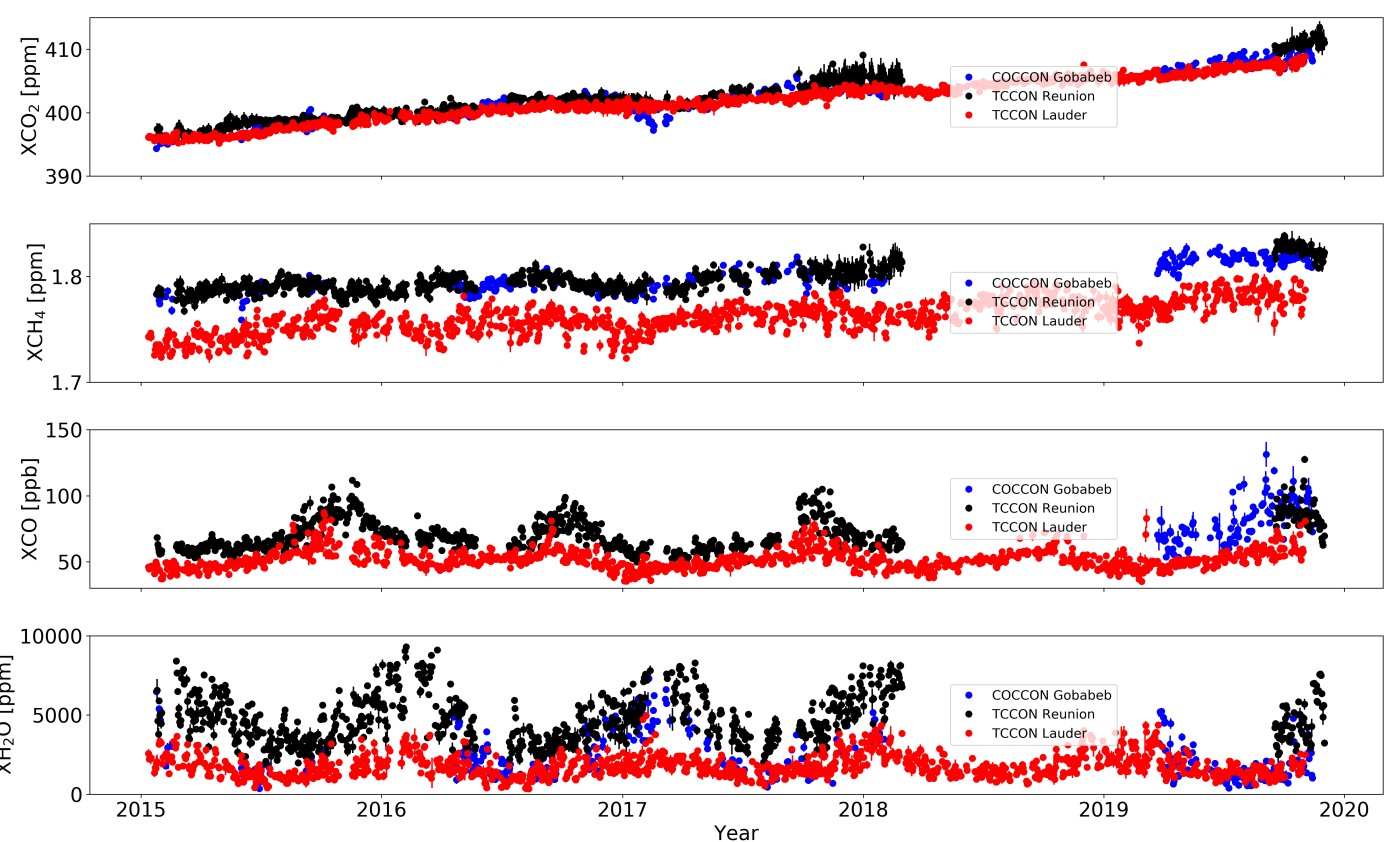

**Figure 4.** Column-averaged dry air mole fraction daily mean time series for $XCO_2$, $XCH_4$, $XCO$ and $XH_2O$ measured at the COCCON site in Gobabeb, Namibia (blue dots) and at the TCCON sites Reunion Island (black dots) and Lauder (red dots). Error bars denote the $1\sigma$ standard deviation of the daily mean values.



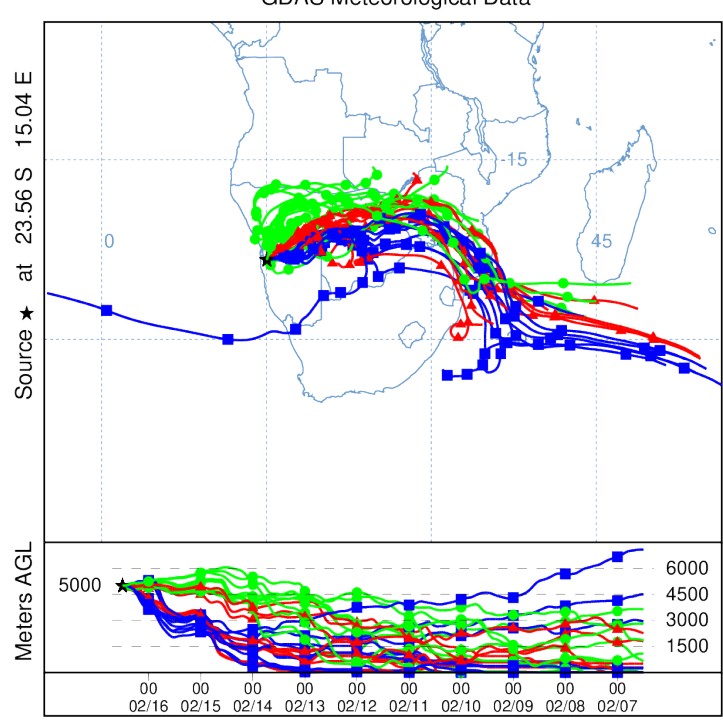

**Figure 5.** NOAA HYSPLIT backward trajectory ensemble simulations on 16 February 2017. The source of the backward trajectories is the COCCON Gobabeb station, 5000 m above ground level.



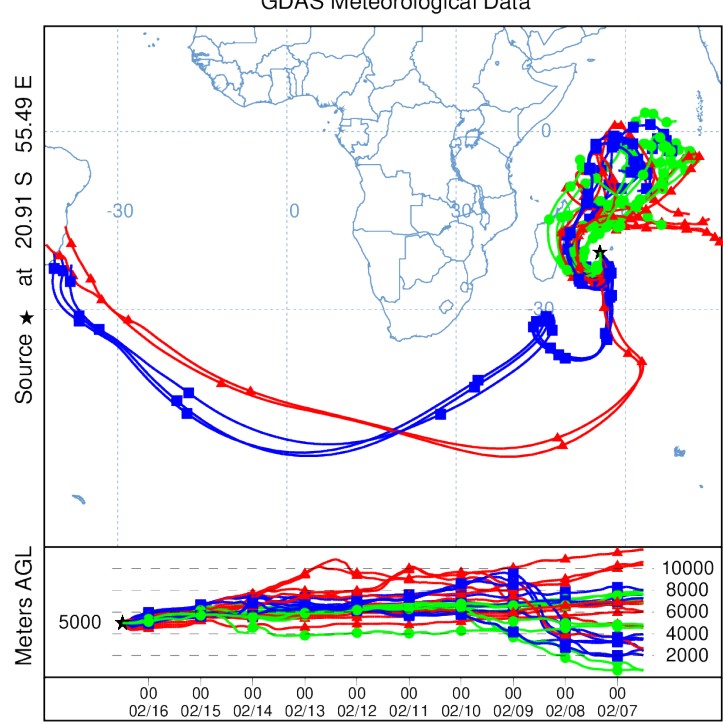

**Figure 6.** Same as Fig. 5, but for the TCCON Reunion Island station.





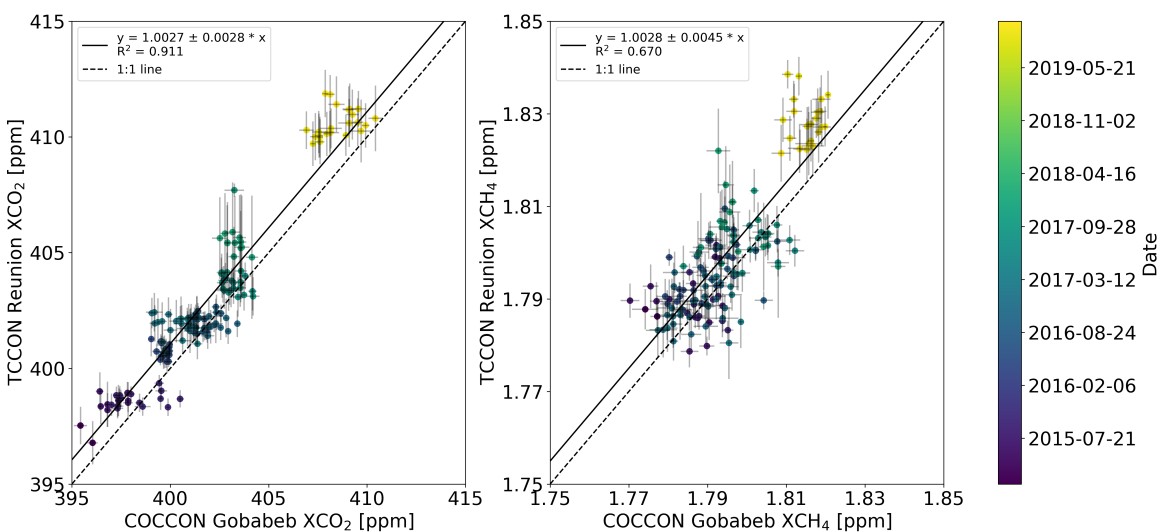

**Figure 7.** Correlation plots between the COCCON Gobabeb and TCCON Reunion Island stations for $XCO_2$ and $XCH_4$ from 2015 to 2019. Shown are daily mean values, errorbars denote the $1\,\sigma$ standard deviation. The colorbar denotes the date of the measurement.

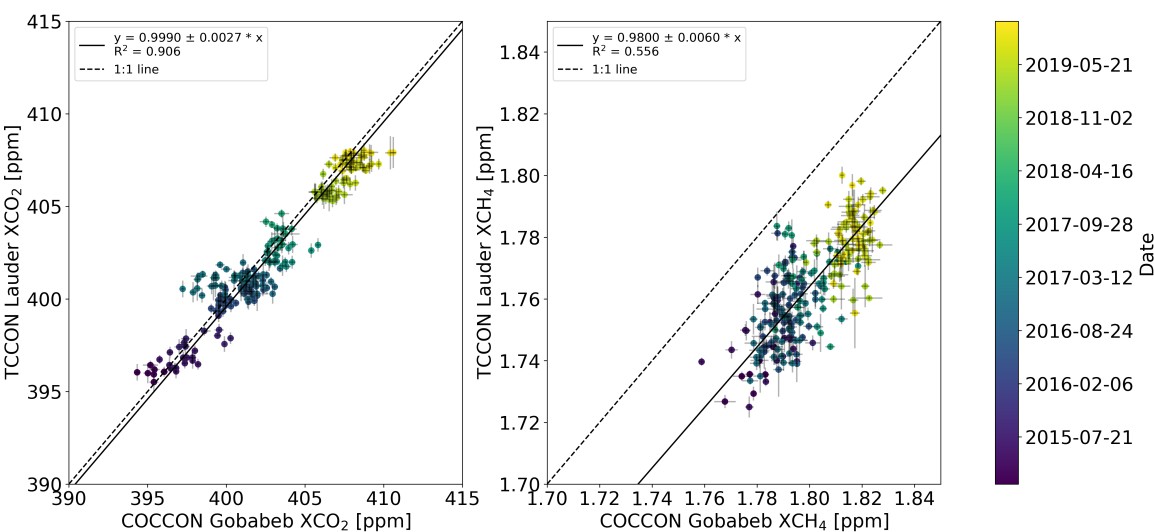

**Figure 8.** Correlation plots between the COCCON Gobabeb and TCCON Lauder stations for $XCO_2$ and $XCH_4$ from 2015 to 2019. Shown are daily mean values, errorbars denote the 1 $\sigma$ standard deviation. The colorbar denotes the date of the measurement.





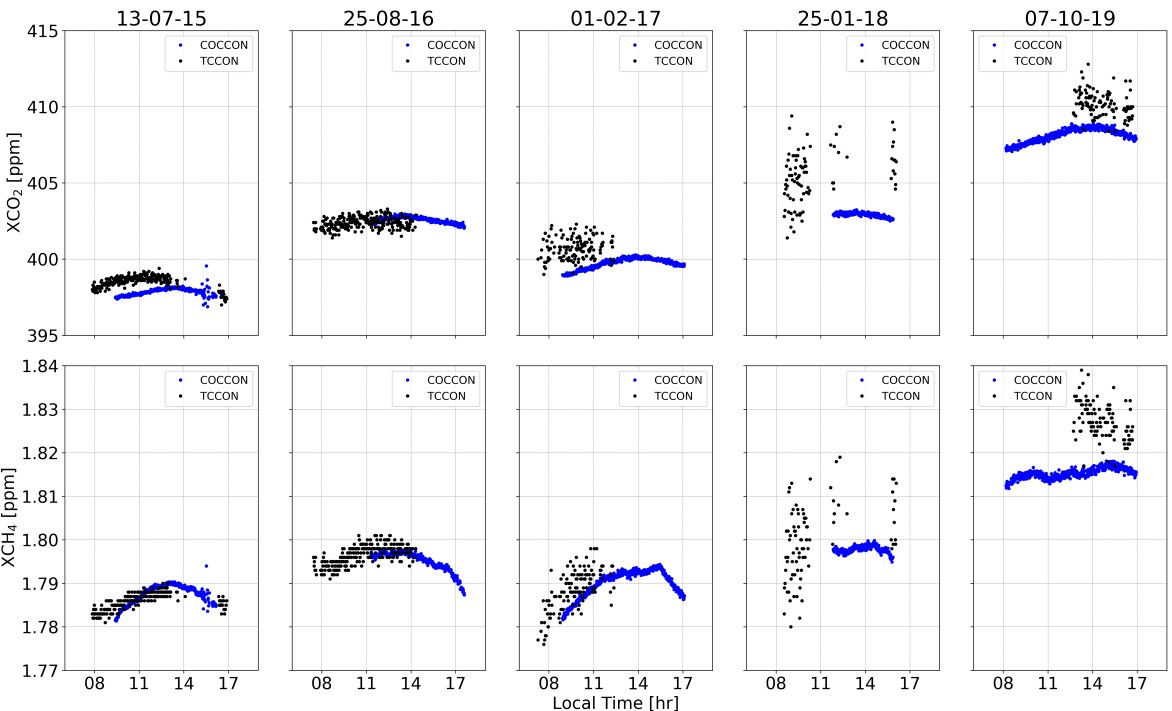

**Figure 9.** Diurnal $XCO_2$ and $XCH_4$ comparisons for one day in 2015, 2016, 2017, 2018 and 2019 between the COCCON station Gobabeb (blue dots) and the TCCON station Reunion Island (black dots).

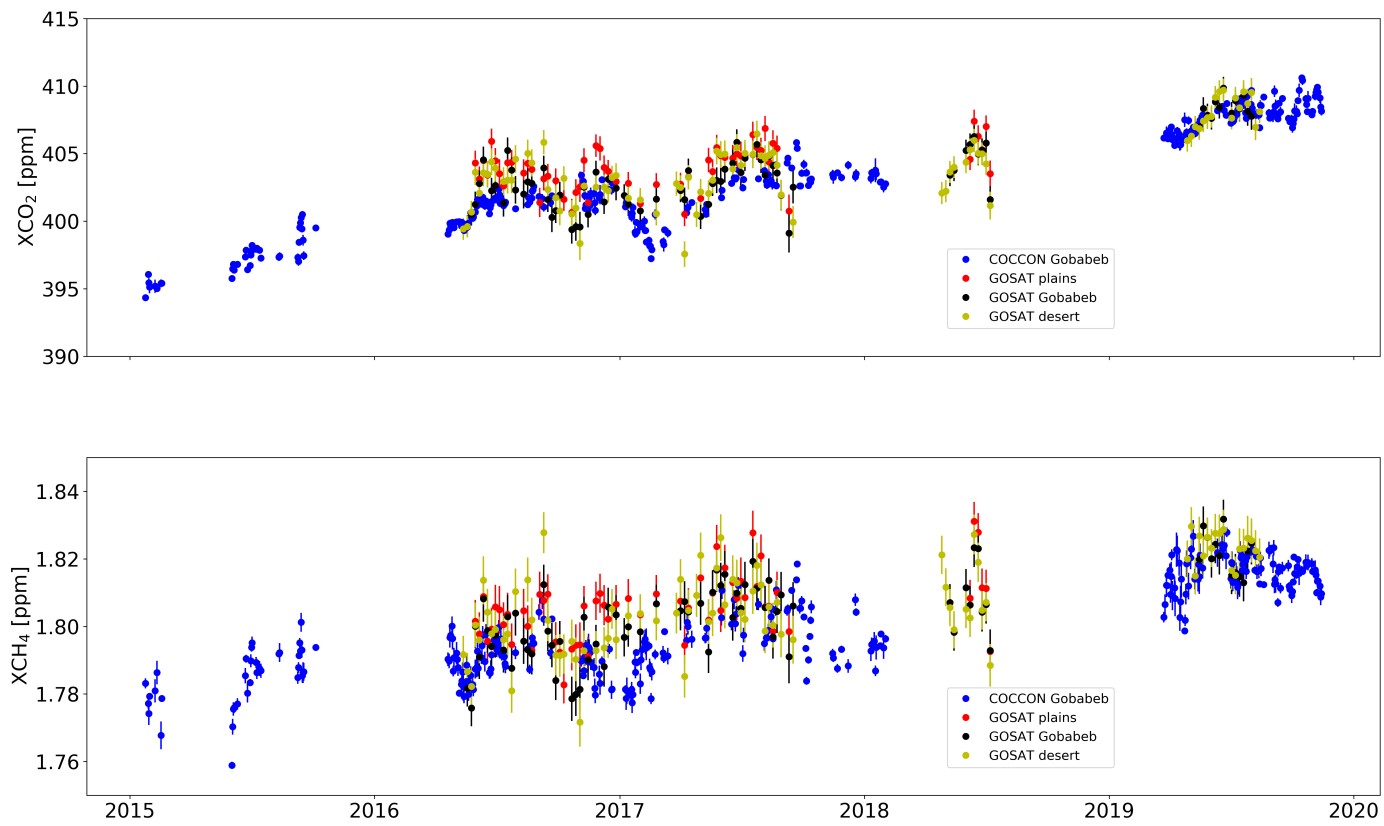

**Figure 10.** Column-averaged dry air mole fraction daily mean time series for $XCO_2$ and $XCH_4$ measured at the COCCON site in Gobabeb (blue dots) and GOSAT observations from the three specific target observation points with different surface albedos close to Gobabeb are shown (red dots: gravel plains, black dots: COCCON site, golden dots: sand desert). Error bars denote the 1 $\sigma$ standard deviation of the daily mean values for COCCON measurements and the measurement error for the GOSAT soundings.

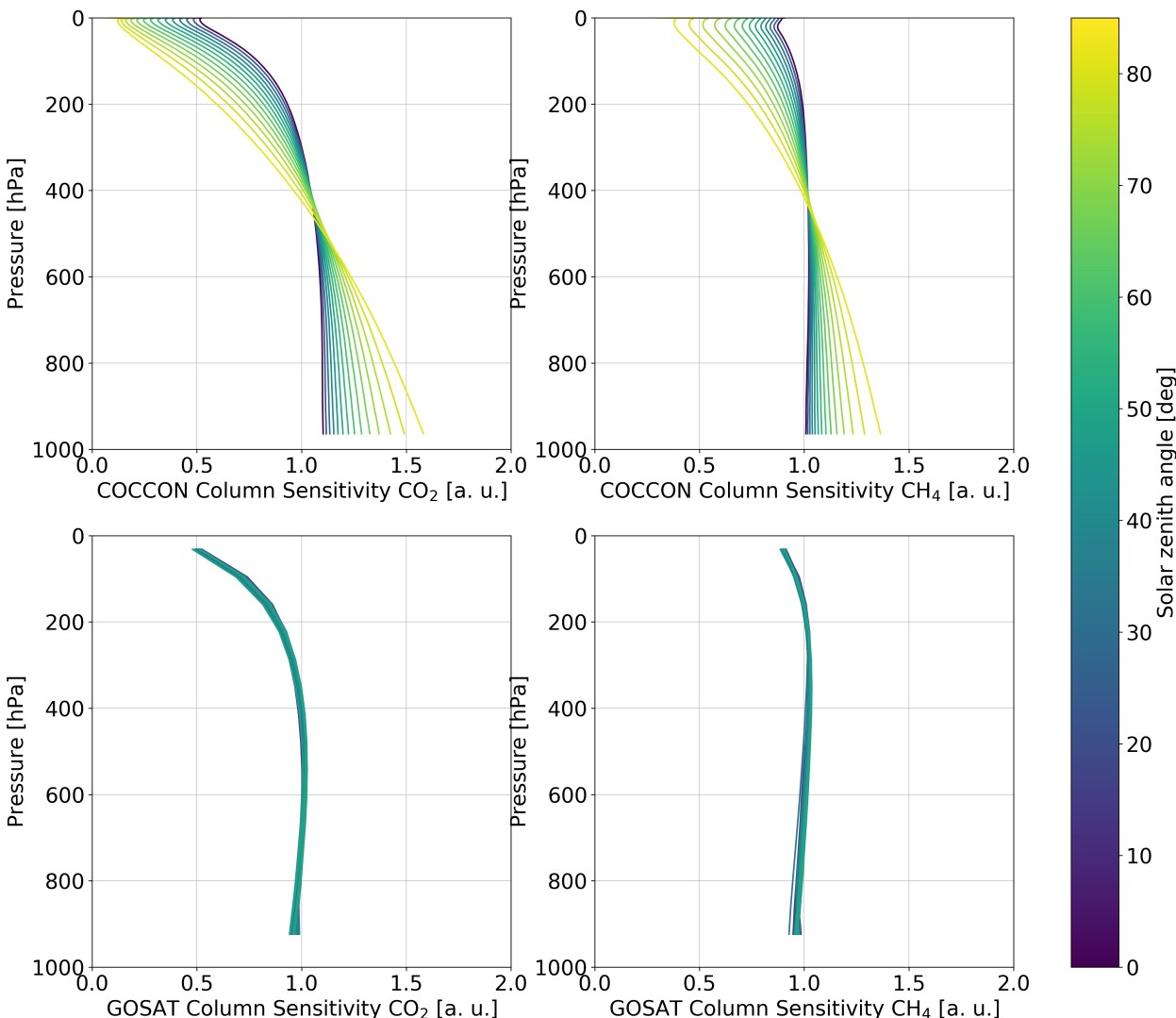

**Figure 11.** $XCO_2$ and $XCH_4$ column averaging kernels for the COCCON Gobabeb and GOSAT observations. The colorbar denotes the SZA. For the COCCON instrument, SZAs from $0°$ to $85°$ are depicted, whereas for GOSAT only the averaging kernels for the actual measurements are shown, with SZAs approximately between $10°$ and $50°$.

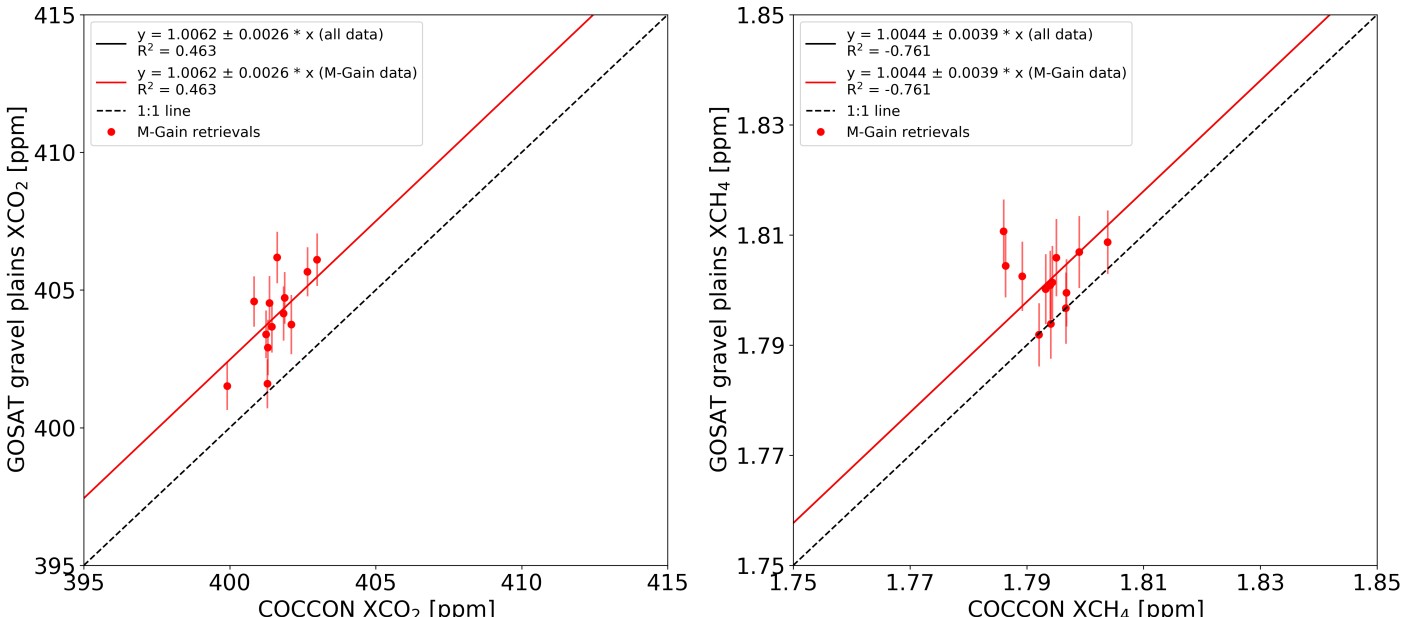

**Figure 12.** Correlation plots between coincident COCCON Gobabeb observations and GOSAT measurements over the gravel plains between 2016 and 2019. For this area GOSAT only performed M-gain soundings (red dots). The red solid line is the best fit line through all M-gain data points. The dotted black line is the 1:1 line. Error bars denote the 1 $\sigma$ standard deviation of the hourly mean values for COCCON measurements and the measurement error for the GOSAT soundings.





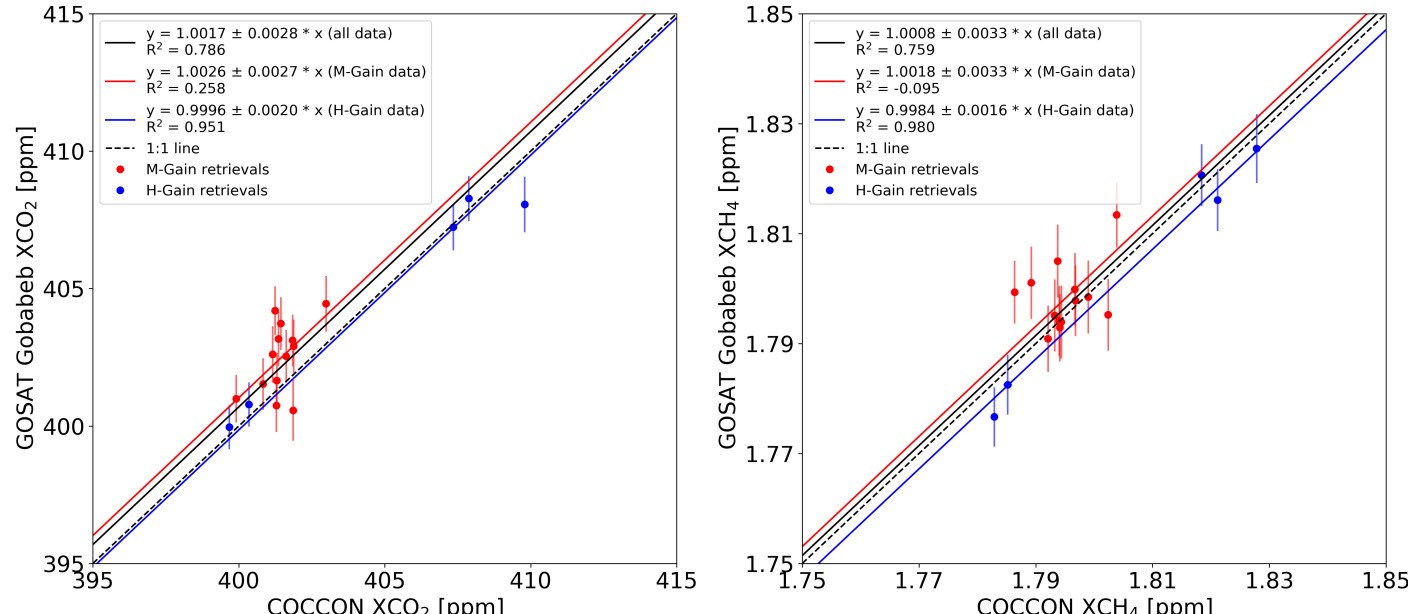

**Figure 13.** Correlation plots between coincident COCCON Gobabeb observations and GOSAT measurements over the COCCON site between 2016 and 2019. For this area GOSAT performed M-gain (red dots) and H-gain (blue dots) soundings. The red solid line is the best fit line through all M-gain data points, the blue solid line is the best fit line through all H-gain data points and the black solid line is the best fit line through all data points. The dotted black line is the 1:1 line. Error bars denote the 1 $\sigma$ standard deviation of the hourly mean values for COCCON measurements and the measurement error for the GOSAT soundings.





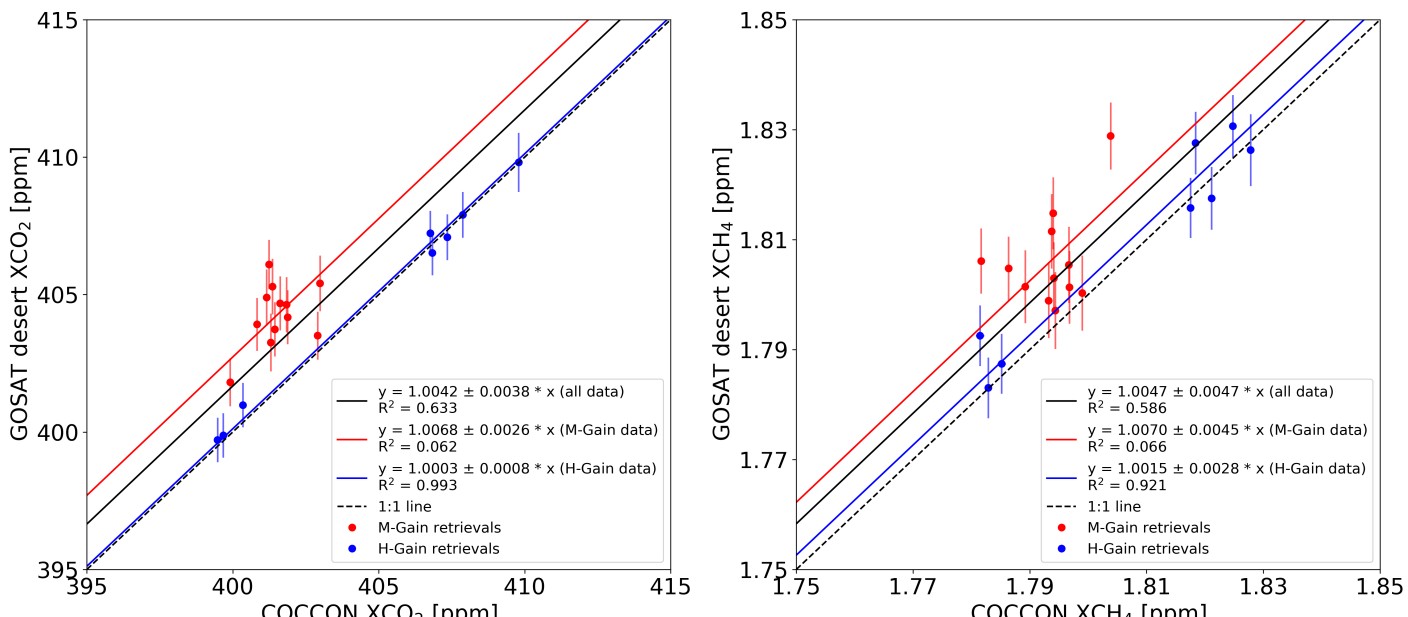

**Figure 14.** Same as Fig. 13, with GOSAT observations over the sand desert.



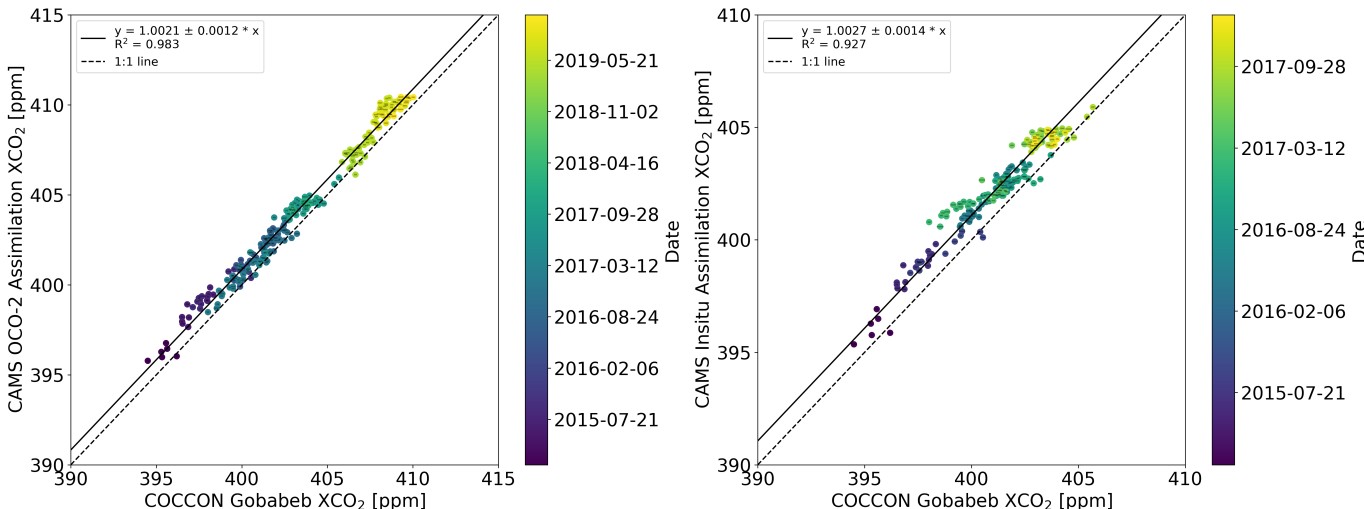

**Figure 15.** $XCO_2$ correlation plots between coincident COCCON Gobabeb observations and CAMS model data. The left panel shows the OCO-2 assimilated model data, the right panel shows the in situ assimilated model data. Note that the OCO-2 assimilated data is available until 2019 and the in situ assimilated data is available until 2018. Error bars denote the $1\,\sigma$ standard deviation of the hourly mean values for COCCON measurements.

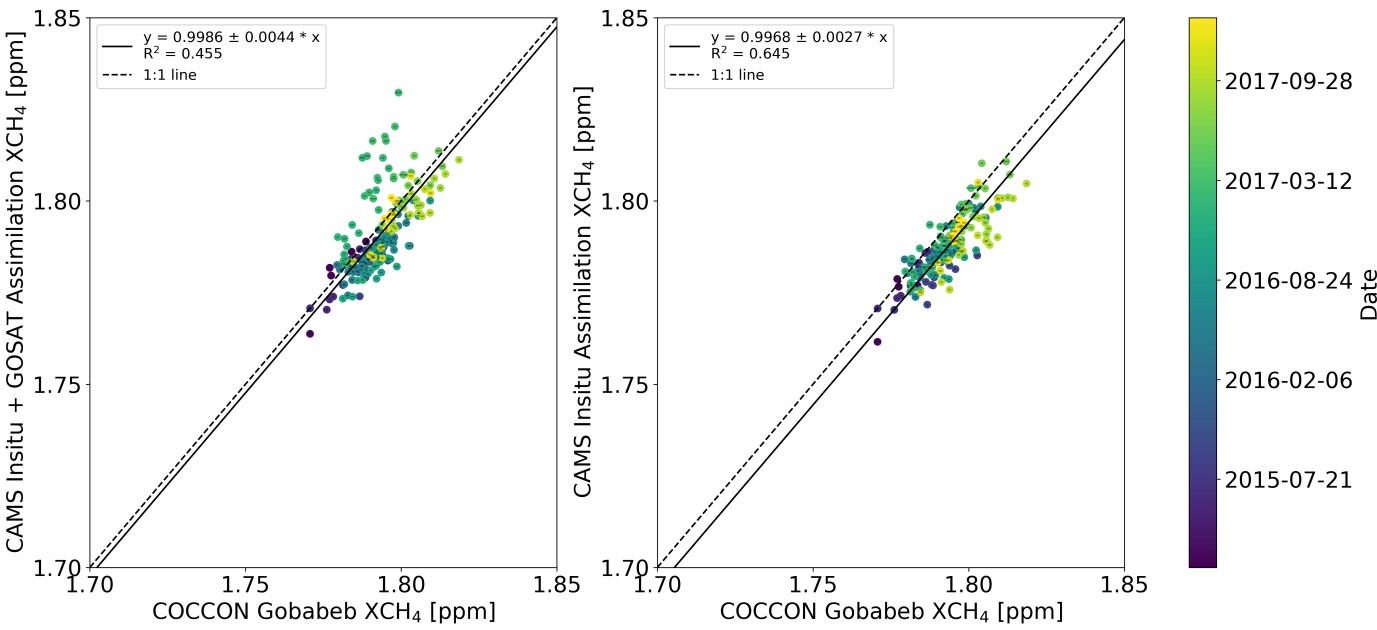

**Figure 16.** $XCH_4$ correlation plots between coincident COCCON Gobabeb observations and CAMS model data. The left panel shows the model data assimilated with in situ and GOSAT data, the right panel shows the in situ assimilated model data. Error bars denote the $1\,\sigma$ standard deviation of the hourly mean values for COCCON measurements.



**Figure A1.** Same as Fig. 2, but for calibration measurements performed between February 2018 and February 2019.