# Peer review of "Long-term column-averaged greenhouse gas observations using a COCCON spectrometer at the high surface albedo site Gobabeb, Namibia"

_Atmospheric Measurement Techniques, 2020_

## Referee Comment (RC2)

**Review of Frey et al, *Long-term column-averaged greenhouse gas observations using a COCCON spectrometer at the high surface albedo site Gobabeb, Namibia**

April 6, 2021

**1 Overview**

It is great to see the start of long-term measurements of ground-based column-averaged greenhouse gases from the African continent; these measurements promise to be a valuable addition for satellite validation and carbon cycle science. The measurements to compare between GOSAT M- and H-gain retrievals are rare, and it is useful to see previously observed biases confirmed. It will be good to see these data made available to the scientific community.

In general, the work presented here is of good quality. While the language could be improved in many places, I don't find that there are many instances where the meaning is unclear and therefore the paper is understandable.

I recommend publication to AMT after addressing a number of changes. None are major, but many are still important to address.

**2 General comments**

- Do you assess the Gobabeb data for potential airmass dependences?

- For the CAMS evaluation, one could argue that restricting based on solar zenith angle rather than time is a better approach. Did you consider this?

- While the paper is understandable, it could be revised to be a bit more clearly written.

- For comparisons, it would also be good to see some plots of the differences as well as the scatter plots or comparative time series.

- The comparisons to Reunion and Lauder TCCON sites are somewhat reassuring, but also complicate matters. This is particularly true for the period where $X_{CO_2}$ disagrees between Gobabeb and Reunion, in early 2017. I find

the postulated explanation and evidence for this unconvincing. I'm not suggesting it isn't the correct reason, just that it's not well-enough supported. I therefore suggest that the authors focus on eliminating instrument problems as a cause of the difference, which should be easy enough given the other gases don't deviate and the $X_{air}$ is stable. Then leave the potential explanation as a hypothesis only and something that warrants further investigation to link it to African biosphere fluxes. Otherwise, at least examination of the CAMS posterior fluxes should be included.

- I missed an explanation of the data gaps early in the time series.

- It might be useful to present time series of the COCCON-CAMS comparisons and their differences.

**3 Specific comments**

- l31: sentence order - switch unprecendented and values

- l32: delete "it was stated" and "that"

- l35: "directly" might be overselling as you would need more than just the measurements themselves, so perhaps delete that word

- l43-44: change to "and CO; however, the measurements ..."

- l52: instrument → network

- l84-85: do you have data to illustrate or support the high albedo?

- l96: "parts ... were"

- l122 - 129: as this is different, it would be good to highlight the differences, and be more specific about the quality control/filtering applied

- l144: good to know there are plans to extend comparisons to other sites

- l149: TCCON doesn't require 0.0035 cm$^{-1}$ spectral resolution, and indeed some instruments don't have that.

- l156: you could update to reflect the Pollard et al (2021) publication comparing the Lauder instruments

- l171: fix location of "e.g." - either inside parentheses with the references, or bring the citations out of the parentheses.

- l181-184: a time-series plot of ME @ Max OPD might be nice to support your statements

- l184: "not self-evident" - please clarify/revise wording.

- l338: while I assume that the bias referred to is between all data and noon-only data, it's not 100% clear, as it could equally apply to bias relative to CAMS. I suggest reworking this sentence: "Although using all COCCON data results in only a small bias of 0.2 ppm for $XCO_2$ and 2 ppb for $XCH_4$ relative to the noon-only data, ..."

- l347: "The absolute values..." - sentence needs revision. "The biases between the CAMS model simulation and the Gobabeb measurements are presented in Table 4." (or something like that)

- l349-350: aren't the end of 2016 and beginning of 2017 the same? I suggest "At the end of 2016 into the beginning of 2017..."

- I suggest combining Tables 2 and 3 to make it easier for the reader to compare between M- and H-gains

- Figure 1 - I have 2 comments. Firstly some broader context would be useful. Secondly, while there are clearly visible differences between the terrain to the NE and SW of the site, it would be nice to have some indicative albedos at the relevant wavelengths.

- Figure 3 - $X_{air}$ is not in arbitrary units - it should be mol mol$^{-1}$ or similar (or even unitless, though mol mol$^{-1}$ is more informative) as the numerator and denominator will both have units of molecules cm$^{-2}$.

- Several locations: data set $\rightarrow$ dataset. Same for the plural

- l379: put "with different surface albedos" before "close to"

- Figure A1 appears to be missing

**4    References**

Pollard, D. F., Robinson, J., Shiona, H. and Smale, D.: Intercomparison of Total Carbon Column Observing Network (TCCON) data from two Fourier transform spectrometers at Lauder, New Zealand, Atmos. Meas. Tech., 14(2), 1501–1510, doi:10.5194/amt-14-1501-2021, 2021.

---

## Author Comment (AC1)

**Authors' answer to the interactive comments of anonymous referee #1 on "Long-term column-averaged greenhouse gas observations using a COCCON spectrometer at the high surface albedo site Gobabeb,Namibia" by Frey et al., Atmos. Meas. Tech. Discuss., amt-2020-444**

First of all, we would like to thank the anonymous referee #1 for the help in further improving the manuscript by a thorough assessment with regards to content and the careful technical proofreading resulting in the identification of several imprecisions. Referee comments are in *blue italic*, responses in black. Changes in the manuscript are in *green italic*.

**1 Overview**

*It is great to see the start of long-term measurements of ground-based column-averaged greenhouse gases from the African continent; these measurements promise to be a valuable addition for satellite validation and carbon cycle science. The measurements to compare between GOSAT M- and H-gain retrievals are rare, and it is useful to see previously observed biases confirmed. It will be good to see these data made available to the scientific community.*
*In general, the work presented here is of good quality. While the language could be improved in many places, I don't find that there are many instances where the meaning is unclear and therefore the paper is understandable.*
*I recommend publication to AMT after addressing a number of changes.*
*None are major, but many are still important to address.*

We thank the referee for this positive feedback and we will address the comments given below. We also appreciate the patience of the referee concerning the language-related issues of the manuscript and the corrections provided.

**2 General comments**

*Do you assess the Gobabeb data for potential airmass dependences?*

In Figure 9 of the original manuscript we show the intraday variability of several days of measurements from the COCCON station in Gobabeb and the closest TCCON station, Reunion Island. We discuss that a small residual airmass dependency is apparent for $XCO_2$ and $XCH_4$. The magnitude of this airmass dependency is similar for the COCCON instrument and the TCCON instrument, which we consider best available reference for ground-based remote sensing.

*For the CAMS evaluation, one could argue that restricting based on solar zenith angle rather than time is a better approach. Did you consider this?*

When we only take the noon data into account, we effectively also restrict the SZA to values between 0° (SZA in Southern hemispheric summer) and 50° (SZA in Southern hemispheric winter). When taking all hourly data into account, we allow SZAs from 0° to 80° (we only evaluate COCCON data with SZAs up to 80° - this selection is included for limiting possible airmass dependent artifacts, which might be introduced by the data analysis or the measurement process itself). Note, that due to the low latitude of the site, the solar elevation is changing much faster during the day than in temperate latitudes. The sun travels through the elevation range between 80° and 50° SZA in just about two hours. Therefore, in our opinion the approach to restrict the comparison to specific SZAs is almost similar to our approach of restricting the comparison based on time. As we describe in section 4.3, the difference between COCCON and CAMS when considering all hourly data and only using the data from local noon is 0.2 ppm for $XCO_2$ and 2 ppb for $XCH_4$.

*While the paper is understandable, it could be revised to be a bit more clearly written.*

We revised the paper and eliminated imprecisions. We especially restructured section 4.1 to focus only on the COCCON/TCCON comparison. We now discuss the origins of the low $XCO_2$ values observed in Gobabeb in a separate section.

*For comparisons, it would also be good to see some plots of the differences as well as the scatter plots or comparative time series.*

We added difference plots to the COCCON TCCON comparisons:

Figure R1: Column-averaged dry air mole fraction daily mean time series for $XCO_2$, $XCH_4$, XCO and $XH_2O$ measured at the COCCON site in Gobabeb, Namibia (blue dots) and at the TCCON sites Reunion Island (black dots) and Lauder (red dots). Error bars denote the 1 σ standard deviation of the daily mean values. *Additionally, the difference $XCO_2$, $XCH_4$, XCO and $XH_2O$ timeseries between Gobabeb and Reunion Island (black dots) and between Gobabeb and Lauder (red dots) are shown in separate panels.*

[Figure]

The comparisons to Reunion and Lauder TCCON sites are somewhat reassuring, but also complicate matters. This is particularly true for the period where $X_{CO_2}$ disagrees between Gobabeb and Reunion, in early 2017. I find the postulated explanation and evidence for this unconvincing. I'm not suggesting it isn't the correct reason, just that it's not well-enough sup-ported. I therefore suggest that the authors focus on eliminating instru-ment problems as a cause of the difference, which should be easy enough given the other gases don't deviate and the $X_{air}$ is stable. Then leave the potential explanation as a hypothesis only and something that warrants further investigation to link it to African biosphere fluxes. Otherwise, at least examination of the CAMS posterior fluxes should be included.

As requested by the referee, we further evidence that the differences seen in early 2017 are very likely not due to instrumental issues. Additionally, as suggested by the referee we now also provide CAMS posterior fluxes to further back up our hypothesis that the differences are linked to the African biosphere. As mentioned above, we restructured this section. In section 4.1 we now focus on the COCCON/TCCON comparison. We added section 5, where we elaborate on our hypothesis about the origin of the low $XCO_2$ values observed in Gobabeb:

*From the end of 2016 until the beginning of 2017, the $XCO_2$ values at the COCCON station at Gobabeb were significantly lower compared to the TCCON stations Reunion Island and Lauder, see section 4.1. We rule out instrumental problems as the reason, as $X_{air}$ is stable and the other observed gases do not show abnormal variations during this period. In order to investigate whether the drawdown of $XCO_2$ at the beginning of 2017 at the Gobabeb station is linked to the African biosphere, in Fig. R2 we present global OCO-2 assimilated CAMS a posteriori surface carbon fluxes for 16 February 2017 12 UTC, the day with the lowest $XCO_2$ values in 2017. We find that in the direct vicinity of Gobabeb, no strong negative carbon fluxes are apparent. From this, we deduce that air parcels with low $CO_2$ concentrations are transported to Gobabeb from other regions of the African mainland with negative surface fluxes. We therefore expect that the drawdown of $XCO_2$ is driven by low $CO_2$ concentrations in higher layers of the atmosphere that are representative for medium- or long-range transport. This is in agreement with the results of section 4.3, where a comparison between COCCON data with CAMS model data shows that the CAMS model version assimilating total column data reproduces the $XCO_2$ drawdown, in contrast to the version assimilating in situ data only. We grant the possibility that the discrepancy between the different CAMS products could also stem from imperfections of the CAMS model.*
*In Fig. 5, we show 10-day backward trajectory ensemble simulations from the National Oceanic and Atmospheric Administration (NOAA) HYSPLIT model (Stein et al., 2016) for 16 February 2017. Initial 3-hourly meteorological input data is provided by the NCEP Global Data Assimilation System (GDAS) model on a 1 degree latitude-longitude grid. The end point of the trajectory analysis is chosen at a height of 5000 m above ground level. All trajectories exhibit a long dwell time over the African continent in regions with strong negative carbon surface fluxes, see Fig. R2. This corroborates the conjecture that the low $XCO_2$ values at Gobabeb are due to the influence of the African biosphere. Most of the trajectories that arrive at 5000 m height at Gobabeb originate from significantly lower levels of the atmosphere, close to the surface, and are then uplifted, as can be seen in the lower panel of Figure 5.*
*In contrast, the backward trajectories for Reunion Island shown in Fig. 6 dwell almost exclusively over the ocean. In Fig. R3 we additionally provide backward trajectories for Gobabeb ending at 1000 m above ground level. In contrast to the trajectories at 5000 m, these originate from the ocean.*

*Figure R2: Global map showing OCO-2 assimilated CAMS a posteriori surface carbon fluxes for 16 February 2017 12 UTC.*

[Figure]

*Figure R3: NOAA HYSPLIT backward trajectory ensemble simulations on 16 February 2017. The endpoint of the backward trajectories is the COCCON Gobabeb station, 1000 m above ground level. The colors and symbols are used to make the different trajectories of the ensemble distinguishable.*

[Figure]

I missed an explanation of the data gaps early in the time series.

The data gap between February and May 2015 was due to software problems. It took so long to figure this out because at the start of the collaboration, there were communication problems between Karlsruhe and Gobabeb. The data gap between October 2015 and April

2016 was induced by customs regulations and associated delays. Therefore, we had to ship the instrument back to Karlsruhe temporarily. We added this information in the text:

*Between February and May 2015, no measurements could be performed due to software problems. In October 2015, the spectrometer was shipped back to Karlsruhe due to custom issues. Observations in Gobabeb were continued from April 2016. In February 2018 the spectrometer was shipped to Karlsruhe for the dual channel upgrade. COCCON measurements were restarted in March 2019.*

It might be useful to present time series of the COCCON-CAMS comparisons and their differences.

We included comparative time series between COCCON and the different CAMS datasets in section 4.3.

*Figure R4: Column-averaged dry air mole fraction daily mean time series for XCO$_2$ and XCH$_4$ at Gobabeb, Namibia. COCCON measurements are shown as blue dots, CAMS model data as red and black dots. For COCCON, we show hourly pooled data, for CAMS we show 3-hourly model output for XCO$_2$ and 6-hourly model output for XCH$_4$.*

[Figure]

[Figure]

**3 Specific comments**

l31: sentence order - switch unprecedented and values

Done

Done

l35: "directly" might be overselling as you would need more than just the measurements themselves, so perhaps delete that word

Ok, done

Ok

Ok

l84-85: do you have data to illustrate or support the high albedo?

We added albedo information from the GOSAT retrievals and the following text:

*Gobabeb is a high albedo station, with a surface albedo derived from GOSAT retrievals at 1.6 µm of 0.4 for the sand desert and 0.45 for the gravel plains.*

Done

l122 - 129: as this is different, it would be good to highlight the differences, and be more specific about the quality control/filtering applied

The analysis is different as we use a new FORTRAN based preprocessing application instead of the previously used python tool. The new source-open tool has been developed in the framework of the ESA project COCCON-PROCEEDS to serve the needs of the COCCON community. Furthermore, we use the recently developed PROFFAST retrieval algorithm, instead of PROFFIT. This code is also source-open and has been created for efficient processing of large numbers of spectra, as the cadence of measurements achieved with low-resolution spectrometers is very high. We expanded our explanation, added a table with the quality filters and reworded as follows:

*Several quality filters are applied, for example requiring a minimum DC level of 5 % of the maximum detector signal level, restricting the tolerable DC variation to 10 % of the DC level of the measured, checking the centerburst location in the IFG and the centerburst amplitudes of forward and backward scans and the relative amplitude of out-of-band artifacts. Table 1 summarizes all quality filters…*
*…PROFFAST is a source-open code for quantitative trace gas analysis, mainly intended for the use with low-resolution FTIR spectrometers. Particular attention has been paid to achieve high processing speed without compromising the high level of accuracy required in the analysis of column-averaged greenhouse gas abundances. For achieving this goal, several measures are taken: (1) PROFFAST uses daily precalculated and tabulated molecular cross-sections derived from line-by-line calculations. (2) Instead of storing the cross sections per discrete layers, the cross-sections are expanded as function of solar*

*zenith angle (SZA), which allows downsizing of the lookup tables by a factor of about five and accelerating the subsequent calculation of atmospheric spectral transmission as function of SZA. (3) The process of convolution of the monochromatic spectrum with the instrumental line shape (ILS) is formulated as a two-step procedure, the first step thins the spectral grid before the convolution is performed.  (4) The state vector of the previous solution is maintained for fitting the next spectrum, as typically the atmospheric variations from spectrum to spectrum are rather small. This strategy allows reducing the number of required iterations to typically two. (5) PROFFAST provides averaging kernels not for each measurement, but as function of a set of SZA values for each measurement day.*
*…*

*Table 1*

| Q1 | *Check of DC level as fraction of ADC range, require 0.05* |
|---|---|
| Q2 | *Check maximum variability of DC level (max. 10 % relative variation in interferogram resulting from 10 coadded scans)* |
| Q3 | *Check FWD / BWD centerburst amplitudes (should agree within 5 %)* |
| Q4 | *Check centerburst location in interferogram record* |
| Q5 | *Check relative amplitude of out-of-band artifacts* |
| Q6 | *Check slope, curvature, and change of curvature of phase spectrum* |
| Q7 | *Check spectral calibration based on cross-correlation of spectral structure wrt a wavenumber-calibrated reference spectrum* |
| Q8 | *Compare spectra derived from forward and backward scans* |

l144: good to know there are plans to extend comparisons to other sites

For example, we also now have a long-term time series of COCCON measurements at the TCCON site Tsukuba that will be used in the future.

l149: TCCON doesn't require 0.0035 cm$^{-1}$ spectral resolution, and indeed some instruments don't have that.

We reworded our statement:

*…which is routinely operated at a spectral resolution of 0.02 cm$^{-1}$…*

l156: you could update to reflect the Pollard et al (2021) publication comparing the Lauder instruments

Done

l171: fix location of "e.g." - either inside parentheses with the references, or bring the citations out of the parentheses.

Done

l181-184: a time-series plot of ME @ Max OPD might be nice to support your statements

We added a time series plot of the modulation efficiency.

*Figure R5: Timeseries of the modulation efficiency at MOPD of the EM27/SUN used in this study. ILS measurements were performed during periods when the instrument was in Karlsruhe for maintenance or detector upgrade. Yellow areas denote measurement periods in Gobabeb. The black bar denotes the time of the detector upgrade.*

[Figure]

l184: "not self-evident" - please clarify/revise wording.

We reworded the sentence:

*This high instrumental stability is remarkable considering that between measurements the EM27/SUN was shipped from Karlsruhe to Gobabeb, including airlift and transport by car on bumpy gravel roads.*

l338: while I assume that the bias referred to is between all data and noon-only data, it's not 100% clear, as it could equally apply to bias relative to CAMS. I suggest reworking this sentence: "Although using all COCCON data results in only a small bias of 0.2 ppm for $XCO_2$ and 2 ppb for $XCH_4$ relative to the noon-only data, ..."

Thanks. We reworded as suggested.

l347: "The absolute values..." - sentence needs revision. "The biases between the CAMS model simulation and the Gobabeb measurements are presented in Table 4." (or something like that)

We reworded the sentence accordingly.

l349-350: aren't the end of 2016 and beginning of 2017 the same? I suggest "At the end of 2016 into the beginning of 2017..."

This is true. We reformulated as suggested.

I suggest combining Tables 2 and 3 to make it easier for the reader to compare between M- and H-gains

Ok, done

Figure 1 - I have 2 comments. Firstly some broader context would be useful. Secondly, while there are clearly visible differences between the terrain to the NE and SW of the site, it would be nice to have some indicative albedos at the relevant wavelengths.

We added a world map showing the COCCON and TCCON station used in this study. We also added information about the albedo in the text:

*Gobabeb is a high albedo station, with a surface albedo derived from GOSAT retrievals at 1.6 µm of 0.4 for the sand desert and 0.45 for the gravel plains…*

*…In Fig. R6 we show the COCCON Gobabeb station in a broader context on a global map together with the TCCON Reunion Island and Lauder stations used in this study.*

*Figure R6: Global map showing the COCCON Gobabeb, TCCON Reunion Island and Lauder sites used in this study.*

[Figure]

Figure 3 – Xair is not in arbitrary units - it should be mol mol$^{-1}$ or similar (or even unitless, though mol mol$^{-1}$ is more informative) as the numerator and denominator will both have units of molecules cm$^{-2}$.

Thanks. We corrected this mistake.

Several locations: data set → dataset. Same for the plural

Done

l379: put "with different surface albedos" before "close to"

Ok

Figure A1 appears to be missing

No, Figure A1 is at the end of the manuscript.

**4 References**

Pollard, D. F., Robinson, J., Shiona, H. and Smale, D.: Intercomparison of Total Carbon Column Observing Network (TCCON) data from two Fourier transform spectrometers at Lauder, New Zealand, Atmos. Meas. Tech., 14(2), 1501–1510, doi:10.5194/amt-14-1501-2021, 2021.

---

## Author Comment (AC2)

**Authors' answer to the interactive comments of anonymous referee #2 on "Long-term column-averaged greenhouse gas observations using a COCCON spectrometer at the high surface albedo site Gobabeb,Namibia" by Frey et al., Atmos. Meas. Tech. Discuss., amt-2020-444**

First of all, we would like to thank the anonymous referee #2 for the help in further improving the manuscript by a thorough assessment with regards to content and the careful technical proofreading resulting in the identification of several imprecisions. Referee comments are in *blue italic*, responses in black. Changes in the manuscript are in *green italic*.

*The manuscript describes a new greenhouse gas observation site with a portable FTIR instrument in a remote site in Namibia. The site gives an opportunity to monitor GHG levels on the African main land. In addition, the high surface albedo of the site and the abrupt change in the albedo near the observation site makes the location a unique opportunity for satellite validations.*
*The analysis of the GHG seasonal cycles at this site and comparisons with GOSAT observations are very beneficial to improve our understanding of the global carbon cycle. However comparisons performed against TCCON and CAMS assimilation model and the conclusions made could be misleading in some cases.*

*Here are some general comments that I think would help improve the manuscript:*

*Although Figure 1 is helpful to demonstrate the terrain around the measurement site I would suggest adding another figure zoomed out that shows where Gobabeb is located as well as the TCCON stations that are used in this study. This will give some better insight to the readers that are not familiar with the region and the TCCON network.*

As suggested by the referee, we included a figure showing the COCCON station as well as the TCCON stations used in this study.

*Figure R1: Global map showing the COCCON Gobabeb, TCCON Reunion Island and Lauder sites used in this study.*

[Figure]

*It is mentioned that for the processing of EM27/SUN spectra PROFAST is used whereas TCCON uses GGG for the retrievals. Have you tried comparing the EM27/SUN retrievals with GGG and see how much difference it would make in the final results?*

Concerning the suggested comparison with GGG, we would like to point out that the code we use for data analysis has already been systematically compared to results collected by collocated TCCON measurements and with Aircore launches in the framework of the FRM4GHG project (Sha et al., 2019) and excellent agreement has been demonstrated. Furthermore, in that publication a comparison between PROFFAST and GGG has been presented. Especially for XCH$_4$ a significant offset between the low-resolution data generated with GFIT and the TCCON reference data was observed. We attribute this offset to the fact that no calibration of GGG for low-resolution applications has been established yet, while clear indications of biases between GGG results (of co-located observations) have been found when applied to high and low-resolution spectra (Hedelius et al., 2017). An offset close to the one found by Sha et al., 2019 was also observed by Ohyama et al, 2020 when they compared EM27/SUN measurements processed with GGG to aircraft measurements. So, without the explicit prior specific determination of AIDCFs and ADCFs for applying GGG on low-resolution spectra the value of such a comparison seems limited. Even if a general set of empirical AIDCFs and ADCFs would be made available, the results of the specific characterization of ILS parameters performed for each COCCON spectrometer such as the instrument operated in Gobabeb still cannot be applied in the current version of GGG. Therefore, we agree that further comparison work on the codes is indeed desirable, but this considerable undertaking is outside the scope of this paper presenting the Gobabeb results. Such a study should be based on the whole COCCON network and should be performed with the upcoming GGG2020 version once appropriate gas calibration factors for low-resolution spectra have been established and the use of instrument-specific ILS parameters has been implemented in GGG.

*Line 140. It is indicated that the accuracy of pressure measurements is 2-3 hPa. First of all what is the precision of the pressure sensor? Second, 2-3 hPa is a large bias. To put it in perspective TCCON requires an accuracy better than 0.3 hPa. Is it possible to calibrate the pressure sensor against a more accurate sensor to improve the accuracy?*

The manufacturer does not provide an explicit quantification of the pressure sensor's precision. In order to estimate the precision, we take the pressure data from one day of measurements performed in Namibia, we randomly choose 20 June 2017. Then we subtract the moving average (averaging over 50 data points) and assume that the empirical standard deviation of this difference, 0.07 hPa, can be regarded as the precision of the sensor.

Regarding the accuracy of the sensor, we thank the referee for pointing this out. We did in fact calibrate our sensor against a co-located CS 100 pressure sensor from the Southern African Science Service Centre for Climate Change and Adaptive Land Management (SASSCAL) network with a stated accuracy of 1.0 hPa for temperatures between 0 °C and 40 °C and long-term stability better than 0.1 hPa per year. (http://www.sasscalweathernet.org/weatherstat_infosheet_we.php?loggerid_crit=8893). We do not directly utilize the data from the CS 100 pressure sensor for our analysis because only hourly values are available. We added text in the manuscript to clarify this:

*…and a precision of 0.07 hPa. In order to increase the level of accuracy, we calibrate our sensor against a co-located CS 100 pressure sensor from the Southern African Science Service Centre for Climate Change and Adaptive Land Management (SASSCAL) network with a long-term stability better than 0.1 hPa per year and a stated accuracy of 0.5 hPa at 20 °C and 1.0 hPa for temperatures between 0 °C and 40 °C. We do not directly use the data from the CS 100 pressure sensor for our analysis as only hourly data are available (http://www.sasscalweathernet.org/weatherstat_infosheet_we.php?loggerid_crit=8893, last access: 07 May 2021).*

*Line 181. It is mentioned that ILS measurements have been performed seven times since 2014 however the dates are not indicated. It is useful to know if ILS has changed after the upgrade done in 2018 to confirm if the changes in the scaling factors after 2018 are ILS related or not.*

We added a timeseries plot of the modulation efficiency at maximum optical path difference. In order to check if the change in the scaling factors obtained from the side-by-side measurements in 2016 and 2018/2019 stems from the upgrade of the instrument in 2018 we average the modulation efficiencies before the upgrade (0.982) and after the upgrade (0.985). The modulation efficiency increased by 0.3 %, which is within the uncertainty budget of 0.3 % using this method (Frey et al., 2019). Therefore, we conclude that the changes in the instrumental line shape due to the upgrade of the COCCON instrument might contribute to the slightly different scaling factors, but they are not the main reason for the changes. We changed the text accordingly:

*In order to investigate if the difference in the calibration factors is linked to the upgrade of the EM27/SUN in 2018, we average the ME at MOPD obtained from the ILS measurements before (0.982) and after (0.985) the upgrade. The ME increased by 0.3 %, which is within the uncertainty budget of 0.3 % using this method. Therefore, we conclude that the changes in the instrumental line shape due to the upgrade of the COCCON instrument might contribute to the slightly different scaling factors, but they are not the main reason for the changes.*

*Figure R2: Timeseries of the modulation efficiency at MOPD of the EM27/SUN used in this study. ILS measurements were performed during periods when the instrument was in Karlsruhe for maintenance or detector upgrade. Yellow areas denote measurement periods in Gobabeb. The black bar denotes the time of the detector upgrade.*

[Figure]

*Table 1, Figure 7 and 8. The comparison against TCCON sites that are thousands of kilometers apart from the measurement site is a bit misleading. Specifically in Table 1, the term "bias" implies one measurement is closer to the truth. Whereas in reality as you have indicated in the manuscript as well, these sites are probably under the influence of different air masses most of the time.*
*Specially for methane as the total column value highly depends on the tropopause height it's not meaningful to compare total column values at Gobabeb and Lauder as the tropopause height is significantly different at the two sites. Generally speaking, I think the timeseries in Figure 4 is useful to give the reader a sense of the GHG seasonal cycles in the southern hemisphere, but the correlation plots and estimating the biases between the sites in my opinion is misleading and unnecessary.*

We chose to compare our results to the TCCON stations because TCCON is the most reliable reference for ground-based GHG total column measurements. We clearly state at the beginning of this section that it is not a side-by-side comparison. In the revised version of the manuscript we add a world map showing the positions of the COCCON/TCCON stations to further emphasize this point, as was also suggested by the referee. We agree that the sites probe different airmasses most of the time and that the comparison of the absolute values at one point in time is not very meaningful. The long-term trend of GHG concentrations in the southern hemisphere should however be comparable at the three sites. We think that the correlation plots are meaningful to see if there are sudden jumps between the data sets, for example due to instrumental issues. For this reason, we would

like to keep these figures. We added the differences between the sites in the timeseries figure to make it easier to follow the long-term trend. We agree that the term "bias" is misleading in this context, and we replaced it with "difference".

Figure R3: Column-averaged dry air mole fraction daily mean time series for $XCO_2$, $XCH_4$, XCO and $XH_2O$ measured at the COCCON site in Gobabeb, Namibia (blue dots) and at the TCCON sites Reunion Island (black dots) and Lauder (red dots). Error bars denote the 1 σ standard deviation of the daily mean values. *Additionally, the difference $XCO_2$, $XCH_4$, XCO and $XH_2O$ timeseries between Gobabeb and Reunion Island (black dots) and between Gobabeb and Lauder (red dots) are shown in separate panels.*

[Figure]

*Figure 9. This figure is not very informative and as it is mentioned by the authors in the text, the curve shapes do not necessarily represent the diurnal cycle and it most probably is related to retrieval errors due to air mass dependencies, a priori profiles, etc. In my opinion, this figure is also misleading and unnecessary.*

We agree that the information content of this figure is limited and we moved it to the appendix together with the accompanying text. We do not want to omit the figure altogether because we think that it has to be shown that a tiny residual curved shape remains due to the retrieval errors mentioned by the referee. We reworded the figure caption and omitted 'diurnal'.

*Line 260-264. The authors conclude that based on the better agreement between CAMS satellite assimilated data and the COCCON measurements at Gobabeb the origin of the CO2 draw down at the beginning of 2017 is from higher levels in the atmosphere.*
*First, CAMS is a model and there are errors associated with it. So the discrepancy might be due to the in situ CAMS assimilated product that could not capture the seasonal variations of CO2 occurring near the surface. Second, even if we accept the assumption that the draw down might have origins higher in the atmosphere, 5000 m above surface seems a bit too high. It would be worthwhile to investigate the trajectories at lower levels for example 500 m and 1000 m above surface to be able to*

*come to a stronger conclusion. You may take into account the instrument column sensitivity in choosing the vertical layers as well.*

We agree that the discrepancy could also be due to errors in the CAMS in situ assimilated product (an investigation of possible errors of the different CAMS products, however, seems out of the scope of this work). We also agree that the variations of $CO_2$ on the African mainland mainly happen close to the surface. Regarding the altitude of the trajectory ending points, it is important to note that we show backward trajectories. Most of the trajectories that arrive at 5000 m height at Gobabeb originate from significantly lower levels of the atmosphere, close to the surface, and are then uplifted, as can be seen in the lower panel of Figure 5. We added a trajectory plot at 1000 m above surface, and the trajectories in that case originate from the ocean. This supports our hypothesis that the low $XCO_2$ values observed at Gobabeb are due to the influence of the African biosphere and low $CO_2$ concentrations are found in higher layers of the atmosphere, above the boundary layer. To further strengthen this point, we included a Figure showing CAMS a posteriori fluxes for February 16, the day for which we calculated the backward trajectories. We clearly see negative surface fluxes for the area from which the backward trajectories originate that end at Gobabeb in 5000 m height. Although the resolution of the model is coarse, it can be seen that at the Gobabeb station itself, no strong negative fluxes are found. We therefore expect to find the low $CO_2$ concentrations in higher layers of the atmosphere that are representative for medium- or long -range transport. We restructured the manuscript. Section 4.1 focuses now solely on the TCCON/COCCON comparison and we added section 5, where we elaborate on our hypothesis for the origin of the $CO_2$ drawdown:

*From the end of 2016 until the beginning of 2017, the $XCO_2$ values at the COCCON station at Gobabeb were significantly lower compared to the TCCON stations Reunion Island and Lauder, see section 4.1. We rule out instrumental problems as the reason, as $X_{air}$ is stable and the other observed gases do not show abnormal variations during this period. In order to investigate whether the drawdown of $XCO_2$ at the beginning of 2017 at the Gobabeb station is linked the African biosphere, in Fig. R4 we present global OCO-2 assimilated CAMS a posteriori surface carbon fluxes for 16 February 2017 12 UTC, the day with the lowest $XCO_2$ values in 2017. We find that in the direct vicinity of Gobabeb, no strong negative carbon fluxes are apparent. From this, we deduce that air parcels with low $CO_2$ concentrations are transported to Gobabeb from other regions of the African mainland with negative surface fluxes. We therefore expect that the drawdown of $XCO_2$ is driven by low $CO_2$ concentrations in higher layers of the atmosphere that are representative for medium- or long-range transport. This is in agreement with the results of section 4.3, where a comparison between COCCON data with CAMS model data shows that the CAMS model version assimilating total column data reproduces the $XCO_2$ drawdown, in contrast to the version assimilating in situ data only. We grant the possibility that the discrepancy between the different CAMS products could also stem from imperfections of the CAMS model.*
*In Fig. 5, we show 10-day backward trajectory ensemble simulations from the National Oceanic and Atmospheric Administration (NOAA) HYSPLIT model (Stein et al., 2016) for 16 February 2017. Initial 3-hourly meteorological input data is provided by the NCEP Global Data Assimilation System (GDAS) model on a 1 degree latitude-longitude grid. The end point of the trajectory analysis is chosen at a height of 5000 m above ground level. All trajectories exhibit a long dwell time over the African continent in regions with strong negative carbon surface fluxes, see Fig. R4. This corroborates the conjecture that the low $XCO_2$ values at Gobabeb are due to the influence of the African biosphere. Most of the trajectories that arrive at 5000 m height at Gobabeb originate from significantly lower levels of the atmosphere, close to the surface, and are then uplifted, as can be seen in the lower panel of Figure 5.*

*In contrast, the backward trajectories for Reunion Island shown in Fig. 6 dwell almost exclusively over the ocean. In Fig. R5 we additionally provide backward trajectories for Gobabeb ending at 1000 m above ground level. In contrast to the trajectories at 5000 m, these originate from the ocean.*

*Figure R4: Global map showing OCO-2 assimilated CAMS a posteriori surface carbon fluxes for 16 February 2017 12 UTC.*

[Figure]

*Figure R5: NOAA HYSPLIT backward trajectory ensemble simulations on 16 February 2017. The endpoint of the backward trajectories is the COCCON Gobabeb station, 1000 m above ground level. The colors and symbols are used to make the different trajectories of the ensemble distinguishable.*

[Figure]

*Minor corrections and comments:*
*Line 14. We find a good agreement for the absolute Xgas values and representative*

*diurnal variability [between TCCON and COCCON?!].*

We reworded the sentence:

*We find a good agreement for the absolute X$_{gas}$ values, apart from an expected XCH$_4$ offset between Gobabeb and Lauder due to significantly different tropopause height, as well as representative intraday variability between TCCON and COCCON.*

*Lines 57-62. The sentence is very long and the reader could be easily lost. Please consider rewording and breaking it up into multiple sentences.*

We split up the sentence.

*Line 67. It seems a sentence is missing here to motivate the current work. What is the mission of this project? i.e. investigate long term stability of EM27/SUNs? or quantifying emission strength from the region of interest or something else?*

We added a sentence motivating this research:

*We demonstrate the excellent long-term stability of the COCCON instrument and its usefulness for satellite and model validation studies. The remainder of this paper is structured as follows.*

*Line 168. For M-gain observations... the sentence is a bit vague. please consider rewording.*

We expanded the explanation:

For M-gain observations other validation sites with ground-based FTIR measurements are sparse. *Yoshida et al. (2013) found no suitable TCCON site for the validation of GOSAT M-gain observations. In recent years, GOSAT M-gain soundings are mainly compared to the Edwards TCCON station, which was established 2013. More recently, Velazco et al. (2019) performed a campaign to validate GOSAT in central Australia using an EM27/SUN.*

*Figure 5 and 6. What do different colors represent?*

The purpose of the different colors and symbols is to guide the eye and make the individual backward trajectories of the ensemble distinguishable. We added this information in the figure caption.

**References**

Sha, M. K., De Mazière, M., Notholt, J., Blumenstock, T., Chen, H., Dehn, A., Griffith, D. W. T., Hase, F., Heikkinen, P., Hermans, C., Hoffmann, A., Huebner, M., Jones, N., Kivi, R., Langerock, B., Petri, C., Scolas, F., Tu, Q., and Weidmann, D.: Intercomparison of low- and high-resolution infrared spectrometers for ground-based solar remote sensing measurements of total column concentrations of $CO_2$, $CH_4$, and CO, Atmos. Meas. Tech., 13, 4791–4839, https://doi.org/10.5194/amt-13-4791-2020, 2020

Hedelius, J. K., Parker, H., Wunch, D., Roehl, C. M., Viatte, C., Newman, S., Toon, G. C., Podolske, J. R., Hillyard, P. W., Iraci, L. T., Dubey, M. K., and Wennberg, P. O.: Intercomparability of $X_{CO2}$ and $X_{CH4}$ from the United States TCCON sites, Atmos. Meas. Tech., 10, 1481–1493, https://doi.org/10.5194/amt-10-1481-2017, 2017

Ohyama, H., Morino, I., Velazco, V. A., Klausner, T., Bagtasa, G., Kiel, M., Frey, M., Hori, A., Uchino, O., Matsunaga, T., Deutscher, N. M., DiGangi, J. P., Choi, Y., Diskin, G. S., Pusede, S. E., Fiehn, A., Roiger, A., Lichtenstern, M., Schlager, H., Wang, P. K., Chou, C. C.-K., Andrés-Hernández, M. D., and Burrows, J. P.: Validation of $XCO_2$ and $XCH_4$ retrieved from a portable Fourier transform spectrometer with those from in situ profiles from aircraft-borne instruments, Atmos. Meas. Tech., 13, 5149–5163, https://doi.org/10.5194/amt-13-5149-2020, 2020